# Determining the effects of pseudouridine incorporation on human tRNAs

Anna D Biela[1], Jakub S Nowak[1], Artur P Biela [1], Sunandan Mukherjee [2], Seyed Naeim Moafinejad [2], Satyabrata Maiti [2], Andrzej Chramiec-Głąbik [1], Rahul Mehta [1,3], Jakub Jeżowski [1], Dominika Dobosz[1], Priyanka Dahate[1], Veronique Arluison [4,5], Frank Wien [6], Paulina Indyka [1,7], Michal Rawski [1,7], Janusz M Bujnicki [2✉], Ting-Yu Lin [1,8✉] & Sebastian Glatt [1,9✉]

## Abstract

**Transfer RNAs (tRNAs) are ubiquitous non-coding RNA molecules required to translate mRNA-encoded sequence information into nascent polypeptide chains. Their relatively small size and heterogenous patterns of their RNA modifications have impeded the systematic structural characterization of individual tRNAs. Here, we use single-particle cryo-EM to determine the structures of four human tRNAs before and after incorporation of pseudouridines (Ψ). Following post-transcriptional modifications by distinct combinations of human pseudouridine synthases, we find that tRNAs become stabilized and undergo specific local structural changes. We establish interactions between the D- and T-arms as the key linchpin in the tertiary structure of tRNAs. Our structures of human tRNAs highlight the vast potential of cryo-EM combined with biophysical measurements and computational simulations for structure-function analyses of tRNAs and other small, folded RNA domains.**

**Keywords** tRNA; Cryo-EM; RNA Modifications; RNA Folding; Molecular Dynamics Simulation
**Subject Categories** RNA Biology; Translation & Protein Quality

## Introduction

tRNAs are the most numerous class of cellular RNAs (Waldron and Lacroute, 1975) and constitute key adaptor molecules in the central dogma of molecular biology that define the genetic code (Crick, 1958; GAMOW, 1954). Each tRNA decodes the sequence of a specific mRNA codon and accurately extends the nascent polypeptide chain using the amino acid carried at its 3′ end (Loveland et al, 2020; Giegé and Eriani, 2023). tRNAs have recently been highlighted as relics of the prebiotic RNA world, and their unique role in defining the RNA-amino acid linkage is a key evolutionary innovation that ushered life into today's RNA-peptide world (Müller et al, 2022). Beyond their canonical role in translation, tRNAs are also involved in a variety of other cellular pathways (Biela et al, 2023). Furthermore, the therapeutic potential of (re)engineered tRNAs for the treatment of genetic diseases has recently gained significant attention (Coller and Ignatova, 2024).

The conserved five-domain architecture of tRNA molecules (Sharp et al, 1985; Holley et al, 1965) includes the amino acid acceptor stem (AAS), the D (dihydrouridine)-loop (or D-arm), the anticodon stem loop (ASL), the variable loop (VL), and the TΨC-arm (T-arm) (Robertus et al, 1974; Kim et al, 1974; Suddath et al, 1974). Despite marked sequence variability, all tRNA molecules not only display a characteristic "cloverleaf"-like secondary structure but also fold into a similar, relatively rigid, L-shaped tertiary structure. This conserved shape allows tRNAs to precisely bridge the distance between mRNA codons in the A/P sites of the ribosome and the peptidyl transferase center (Rodnina and Wintermeyer, 2010). High-resolution structural analyses of tRNAs within ribosomes have been achieved by single particle cryo-EM (Saito et al, 2022; Buschauer et al, 2020; Pochopien et al, 2021) and lately by cryo-ET (Tegunov et al, 2021; Hoffmann et al, 2022; Xue et al, 2022; Robertus et al, 1974). Despite the fact that tRNAs continuously transition between bound and unbound states, very few tRNAs (and only one non-canonical human tRNA (Itoh et al, 2009)) have been structurally characterized without interaction partners (Moras et al, 1980; Woo et al, 1980; Basavappa and Sigler, 1991; Bénas et al, 2000; Kim et al, 1974; Suddath et al, 1974). Required quantities and heterogenous patterns of numerous RNA modifications have impeded the systematic structural characterization of endogenous tRNAs in isolation. Cryo-EM analysis of naked tRNAs and other small RNA molecules has remained a challenge due to their small size, limiting our ability to understand their folding, stability, dynamics, and variability.

[1]Malopolska Centre of Biotechnology, Jagiellonian University, 30-387 Krakow, Poland. [2]Laboratory of Bioinformatics and Protein Engineering, International Institute of Molecular and Cell Biology in Warsaw, 02-109 Warsaw, Poland. [3]Doctoral School of Exact and Natural Sciences, Jagiellonian University, 30-348 Krakow, Poland. [4]Laboratoire Léon Brillouin LLB, UMR12 CEA CNRS, CEA Saclay, 91191 Gif-sur-Yvette, France. [5]Université Paris Cité, UFR Sciences du vivant, 75006 Paris, Cedex, France. [6]Synchrotron SOLEIL, L'Orme des Merisiers, Saint Aubin BP48, 91192 Gif-sur-Yvette, France. [7]National Synchrotron Radiation Centre SOLARIS, Jagiellonian University, 30-392 Krakow, Poland. [8]Department of Biosciences, Durham University, DH1 3LE Durham, UK. [9]University of Veterinary Medicine Vienna, 1210 Vienna, Austria. ✉E-mail: janusz@iimcb.gov.pl; ting-yu.lin@durham.ac.uk; sebastian.glatt@uj.edu.pl

Across the tree of life, tRNAs are known to contain over 100 different modified RNA nucleotides (Suzuki, 2021; Cappannini et al, 2021), which are introduced post-transcriptionally by specialized modifying enzymes (Väre et al, 2017; Cappannini et al, 2021). Modifications can be found at various positions of the nucleotide and in all regions of tRNAs (Jackman and Alfonzo, 2013). The ASL is a modification hotspot (Krutyhołowa et al, 2019), where modifications directly affect codon-anticodon interactions and translation elongation rates (Nedialkova and Leidel, 2015). Modifications in the other domains typically do not directly affect decoding but promote aminoacylation, processing, maturation and correct folding of tRNAs (Lorenz et al, 2017). These processes can be disrupted by various mutations in tRNA modifying enzymes, which are associated with severe human diseases like mitochondrial diseases, diabetes, cancer, intellectual disabilities, Amyotrophic Lateral Sclerosis, Dubowitz-like syndrome, Noonan-like syndrome, familial dysautonomia and ataxia (Torres et al, 2014; Suzuki, 2021; Schaffrath and Leidel, 2017).

Pseudouridine ($\Psi$) (Veerareddygari et al, 2016) is the most abundant modification in tRNAs (Cappannini et al, 2021), but it is also found in many other RNA families (Lin et al, 2021)—therefore, $\Psi$ is sometimes referred to as the 5th RNA nucleotide (Cohn and Volkin, 1951). $\Psi$ is the 5-ribosyl isomer of uridine that allows the uracil base to form an additional water-mediated hydrogen bond with the neighboring nucleotides, resulting in the local stabilization of folded RNA elements (Voegele et al, 2023). As $\Psi$-sites rarely appear in the anticodon or acceptor stem region of human tRNAs (Bare and Uhlenbeck, 1985), it was speculated that they only indirectly affect decoding or aminoacylation (Borchardt et al, 2020). In humans, pseudouridylation on tRNAs is carried out by five "stand-alone" pseudouridine synthases (PUS), which are each responsible for the isomerization of uridines at specific nucleotide positions (Rintala-Dempsey and Kothe, 2017). The appearance of $\Psi$ in many different RNA families and locations has raised the intriguing question whether the modification fulfills the same function everywhere or whether its effect is position- and context-specific (Voegele et al, 2023). The limited structural information on isolated tRNAs has precluded a deeper structure-function level understanding of how modifications alter the folding, dynamics, stability, or conformation of tRNAs. For instance, only tRNA$^{Phe}$ has been imaged by crystallography in its unmodified and modified states (Biela et al, 2023) and NMR has provided us with the structures of small RNA hairpins carrying certain sets of modifications (Vendeix et al, 2008; Denmon et al, 2011). Whereas some tRNAs seem to be translationally active even without modifications (Harrington et al, 1993), it is known that many modifications are essential and mutations in modification enzymes are associated with a variety of severe human diseases (Hawer et al, 2018).

Here, we present single particle cryo-EM structures of various in vitro transcribed human nuclear-encoded tRNAs at overall resolution ranges up to 4.9 Å. By analyzing isolated tRNAs after enzymatic incorporation of specific sets of $\Psi$s at their naturally occurring positions, we show that $\Psi_{13}$ (D-arm) and $\Psi_{55}$ (T-arm) are crucial for the folding of most tRNAs and that these and other $\Psi$ sites result in local, context-dependent structural effects. We performed molecular simulations that corroborate our experimental observations and highlight the importance of interaction between the D- and T-arms for overall tertiary structural stability. Finally, we isolated individual tRNA iso-acceptors from human cells and determined their intermediate-resolution structures by

cryo-EM. Our work provides direct experimental evidence for several fundamental principles that guide tRNA biogenesis and at the same time illustrates the technical feasibility of cryo-EM for studying small RNA molecules.

# Results

## Heterogeneity of folding patterns among human tRNAs

To investigate the structure, folding, and stability of human tRNAs, we produced 10 human tRNA sequences individually by T7 run-off in vitro transcription (IVT; Appendix Fig. S1A,B). We annealed and purified tRNAs (Fig. 1A) to obtain pure and homogenous samples displaying the expected molecular size and similar mobility in a native electrophoresis (Fig. 1B; Appendix Fig. S1C,D). Next, we measured the melting temperature (Tm) of each human tRNA sample by monitoring the increase in fluorescence intensity as RiboGreen™ binds to tRNA regions that become single-stranded upon unfolding under a temperature gradient. While applying a temperature gradient, 8 out of 10 purified transcripts indeed folded into stable conformations that displayed Tm values well above human body temperature (Fig. 1C; Appendix Fig. S1E). tRNA$^{Pro}_{UGG}$ and tRNA$^{Val}_{UAC}$ display very low Tm values and their hydrodynamic radii are the largest compared to others, suggesting they did not fold properly, and they may rely on modifications for folding. We noticed that their hydrodynamic radii are still significantly smaller than the predicted size of a fully unfolded tRNA, indicating partial folding.

To gain a comprehensive understanding of the folding and unfolding mechanisms of a tRNA, we analyzed the behavior of the folded tRNA$^{Gln}_{UUG}$ at different temperatures by monitoring changes in dynamic light scattering (DLS) and in silico simulation (Fig. EV1A,B). DLS of the folded tRNA$^{Gln}_{UUG}$ shows a relatively small increase in the hydrodynamic radius over an increasing thermal gradient, suggesting the presence of remaining structural elements even at very high temperatures. Our molecular simulations show that the conserved canonical $G_{19}$-$C_{56}$ pair at the tip between D- and T-loops represents the molecular "Achilles heel" of tRNAs. Once this contact is lost, the remaining non-canonical interactions between D- and T-loops break easily, further destabilizing the characteristic tRNA tertiary structure (Fig. EV1C and Movie EV1). However, the individual stem structures of the tRNA arms remain to a large extent base-paired even at higher temperatures, suggesting that the observed "unfolding" transition during the thermal ramp is caused by the separation of D- and T-loops (Fig. EV1D,E and Movie EV2). This observation is consistent with chemical probing experiments using bacterial and yeast tRNAs (Yamagami et al, 2022; Wilkinson et al, 2005). We further conclude that GC content is a very weak criterion for predicting the stability of folded RNA domains (Fig. 1C) and the formation of functional tertiary RNA structures seems to instead depend on individual canonical and non-canonical base pairs.

## Single particle cryo-EM structures of unmodified human tRNAs

To characterize the three-dimensional structures of individual human tRNAs, we vitrified purified tRNA$^{Asp}_{GUC}$, tRNA$^{Arg}_{UCU}$,

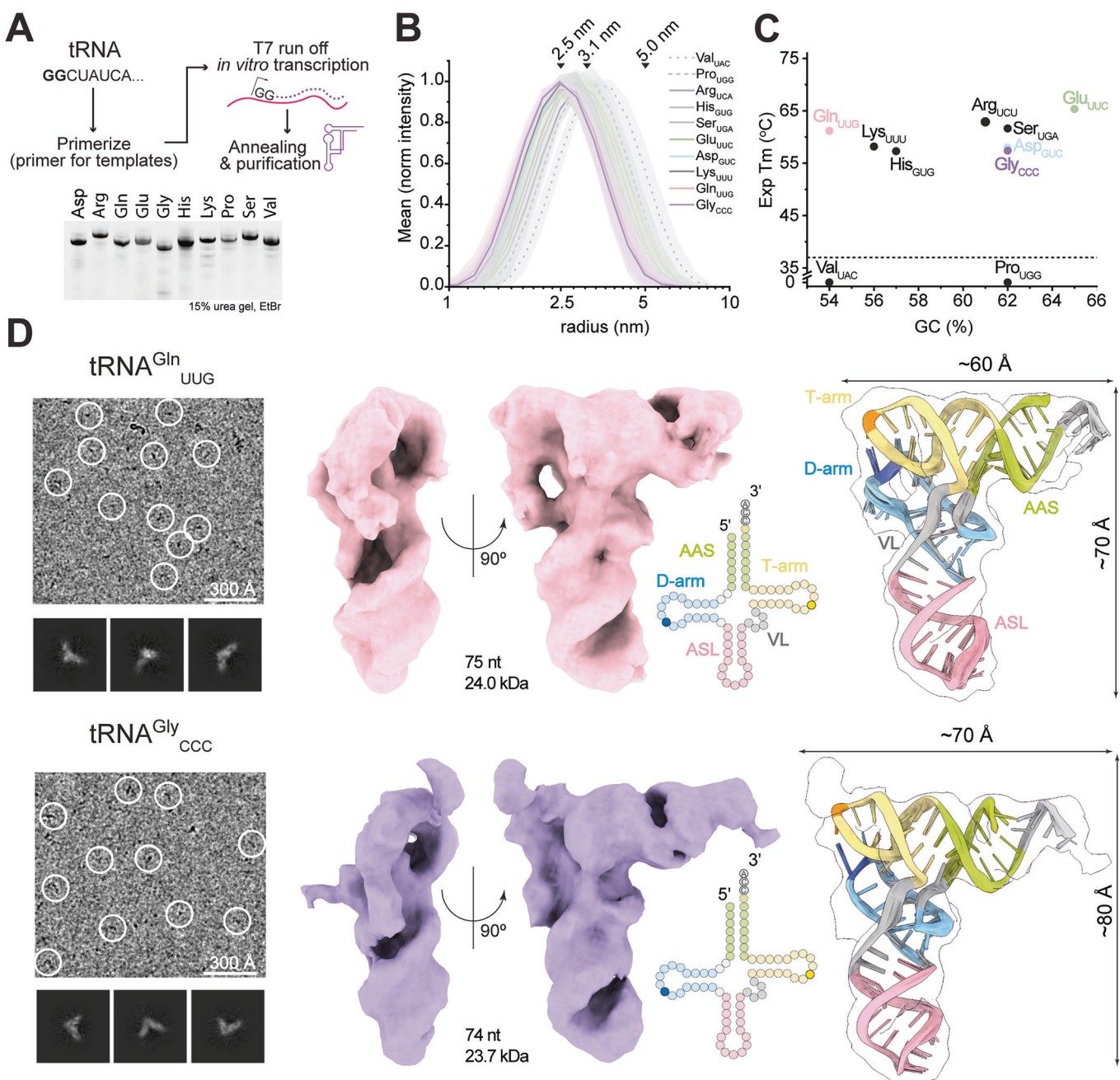

**Figure 1. Cryo-EM structures of in vitro transcribed human tRNAs.**

(A) Top: Scheme of in vitro transcription (IVT) production of human cytosolic tRNAs. DNA templates contain T7 polymerase recognition site and tRNA sequence with 5′ GG. Bottom: Urea gel showing purity of produced tRNAs. (B) nanoDLS determined hydrodynamic radius (rH) profiles of each in vitro transcribed tRNAs. 2.5 nm corresponds to rH of folded tRNA, 3.1 nm corresponds to rH of partially unfolded tRNA, 5.0 nm corresponds to rH of unfolded tRNA. Shaded regions around the curves represent s.d. error. Biological replicates, $N = 3$. (C) Plot of experimentally determined melting temperatures (Exp Tm, °C) of in vitro transcribed tRNAs determined using Quant-iT RiboGreen™ RNA fluorescent probe versus GC content (%). The dashed line corresponds to 37 °C. tRNA$^{Val}_{UAC}$ and tRNA$^{Pro}_{UGG}$ have no inflection point in Tm determination. (D) Representative micrograph and 2D class averages of human tRNA$^{Gln}_{UUG}$ (upper left) and tRNA$^{Gly}_{CCC}$ (lower left). Cryo-EM reconstructions of human tRNA$^{Gln}_{UUG}$ (upper middle) and tRNA$^{Gly}_{CCC}$ (lower middle) with an ensemble of 10 atomic models fit into the density (right). Inset: Clover-leaf secondary structure representation of tRNA highlighted with each domain: acceptor arm (AAS, green); T-arm (yellow) with highlighted C$_{56}$ (dark yellow); D-arm (blue) with highlighted G$_{19}$ (dark blue); variable loop (VL, gray); anticodon arm (ASL, pink). 5′ and 3′ ends are indicated. All cryo-EM maps are contoured to RMSD = 8. Source data are available online for this figure.

**Table 1. Cryo-EM data collection, refinement and validation statistics.**

| | tRNA$^{Gln}_{UUG}$ unmodified EMPIAR 11514 EMDB 16940 | tRNA$^{Gln}_{UUG}$ 3/7 – 16939 | tRNA$^{Gln}_{UUG}$ 4/7 11515 16941 | tRNA$^{Gly}_{CCC}$ unmodified 11516 16943 | tRNA$^{Gly}_{CCC}$ 4/7 11517 16945 | tRNA$^{Gly}_{CCC}$ 3/4/7 – 16946 | tRNA$^{Glu}_{UUC}$ unmodified – 16947 | tRNA$^{Glu}_{UUC}$ 4/7 11518 16948 | tRNA$^{Asp}_{GUC}$ unmodified – 16950 | tRNA$^{Asp}_{GUC}$ 4/7 11519 16951 |
|---|---|---|---|---|---|---|---|---|---|---|
| *Data collection and processing* | | | | | | | | | | |
| Magnification | 175k | 175k | 175k | 175k | 175k | 175k | 175k | 175k | 175k | 175k |
| Voltage (kV) | 300 | 300 | 300 | 300 | 300 | 300 | 300 | 300 | 300 | 300 |
| Electron exposure (e−/Å$^2$) | 40 | 40 | 40 | 40 | 40 | 40 | 40 | 40 | 40 | 40 |
| Defocus range (μm) | −0.8 to −2.4 | −0.8 to −2.4 | −0.8 to −2.4 | −0.8 to −2.4 | −0.8 to −2.4 | −0.8 to −2.4 | −0.8 to −2.4 | −0.8 to −2.4 | −0.8 to −2.4 | −0.8 to −2.4 |
| Pixel size (Å) | 1.72 | 0.86 | 1.692 | 1.72 | 1.72 | 1.72 | 1.72 | 0.86 | 1.692 | 1.692 |
| Symmetry imposed | C1 | C1 | C1 | C1 | C1 | C1 | C1 | C1 | C1 | C1 |
| Initial particle images (no.) | 1,346,327 | 819,259 | 1,500,884 | 699,147 | 914,161 | 789,255 | 584,640 | 784,375 | 602,174 | 644,714 |
| Final particle images (no.) | 304,626 | 48,096 | 306,549 | 153,531 | 220,280 | 186,078 | 66,863 | 38,501 | 85,573 | 81,488 |
| Map resolution (Å) FSC = 0.143 | 5.10 | 4.93 | 5.10 | 5.23 | 5.34 | 5.54 | 5.28 | 4.95 | 5.26 | 5.83 |

tRNA$^{Gln}_{UUG}$, tRNA$^{Glu}_{UUC}$, tRNA$^{Gly}_{CCC}$, tRNA$^{His}_{GUG}$, tRNA$^{Lys}_{UUU}$, tRNA$^{Ser}_{UGA}$ on cryo-EM grids. After grid screening and sample optimization, we collected larger datasets for human tRNA$^{Gln}_{UUG}$ and tRNA$^{Gly}_{CCC}$. After applying basic image correction, particle picking, and classification procedures, we observed numerous 2D classes displaying tRNA-like particles with the expected diameter of approximately 90 Å for human tRNA$^{Gln}_{UUG}$ and tRNA$^{Gly}_{CCC}$ (Fig. EV2; Appendix Fig. S2). Iterative rounds of 3D classification yielded homogenous sets of particles, allowing us to reconstruct cryo-EM maps at nominal FSC$_{0.143}$ resolutions of 5.3 Å and 4.9 Å, respectively (Table 1). Conventional methods of resolution estimation are challenged by very small objects (Wu and Lander, 2020), but the obtained maps clearly resolve distinct structural features (e.g., RNA phosphate-backbone) that indicate resolutions better than 6 Å (Fig. 1D). Each of the structures displays a canonical L-shaped tRNA, wherein the individual domains are distinguishable. In both structures, the ASL region is well defined, and the major and minor grooves of the anticodon and acceptor stems are recognizable. In tRNA$^{Gln}_{UUG}$, the elbow region appears compact and the interaction between the D- (G$_{19}$) and T-loop (C$_{56}$) is visible. In tRNA$^{Gly}_{CCC}$, the CCA at the 3′-end is visible, but the T-arm is less well resolved. Next, we used restrained flexible fitting implemented in SimRNA (Boniecki et al, 2015) to check whether sequence-based models can be reasonably fitted into the maps at the obtained resolution. For both tRNAs, we obtained ensembles of atomic models, which fit well into the cryo-EM maps (Fig. 1D). The top-scoring models for each tRNA are highly similar (Fig. EV3A) and slight variations appear only at the 3′-end and in the anticodon loop. Hence, the combination of structural modeling together with spatial restraints from intermediate resolution cryo-EM maps is sufficient to obtain reliable models of human tRNA$^{Gln}_{UUG}$ and tRNA$^{Gly}_{CCC}$. In summary, we present structural models of two unmodified human tRNAs by combining single-particle cryo-EM and map-assisted semi-supervised modeling, representing the smallest asymmetric particles (~25 kDa) analyzed by cryo-EM to date.

## Site-specific introduction of Ψs affects tRNA stability and structure

We next asked how the introduction of the most common tRNA modification, pseudouridine, influences the structures of tRNAs. Interestingly, the recently developed mRNA vaccines use total substitution of all uridines by Ψ or N1-methyl-Ψ (m$^1$Ψ) (Sahin et al, 2021) to enhance mRNA stability and reduce undesired immunogenic responses (Karikó et al, 2008). However, the mRNA molecules in the vaccines do not need to fold into a defined, stable tertiary structure. We expected that saturating tRNAs with these modifications would negatively affect their structures. To verify this, we synthesized tRNA$^{Gln}_{UUG}$ by IVT with complete replacement of all 19 uridines with either Ψ or m$^1$Ψ. While tRNA$^{Gln}_{UUG}$ was well-folded in its unmodified state (Tm 61.7 °C), its Tm decreased by more than 23 °C (U → Ψ; Tm 38.1 °C) and 29 °C (U → m1Ψ; Tm 32.1 °C) when fully substituted with either Ψ-variant (Fig. 2A). To address the initial question in a biologically meaningful way, we tested whether introducing Ψs exclusively at their five naturally occurring sites in tRNA$^{Gln}_{UUG}$ (Cappannini et al, 2021) would stabilize it. We produced and purified wild-type or catalytic mutants (carrying an Asp-to-Ala (DA) substitution (Huang et al, 1998)) of all five human PUS enzymes expressed in bacteria (PUS1$_{30-427}$, TRUB1/PUS4$_{1-349}$, PUS7$_{98-661}$ and PUS10$_{1-529}$) or insect cells (PUS3$_{1-481}$) (Figs. 2B and EV4A). PUS1, PUS3, TRUB1/PUS4, PUS7, and PUS10 catalyze formation of Ψ$_{27/28}$, Ψ$_{38/39}$, Ψ$_{55}$, Ψ$_{13}$, and Ψ$_{54}$, respectively (Rintala-Dempsey and Kothe, 2017). All of the catalytic mutants bind to tRNAs with comparable affinity to the wild-type versions (Fig. EV4B). Using a N-cyclohexyl-N'-β-(4-methylmorpholinium)ethylcarbodiimide (CMC)-based primer extension assay, we show that the purified PUS enzymes catalyze highly specific pseudouridinylation of their expected target sites while leaving other uridines unmodified (Fig. 2C). Combining all five enzymes into one reaction allowed us to produce tRNA$^{Gln}_{UUG}$ with all known naturally occurring Ψ-sites (Ψ$_{13}$, Ψ$_{28}$, Ψ$_{39}$, Ψ$_{54}$, and Ψ$_{55}$) (Cappannini et al, 2021) in vitro. We

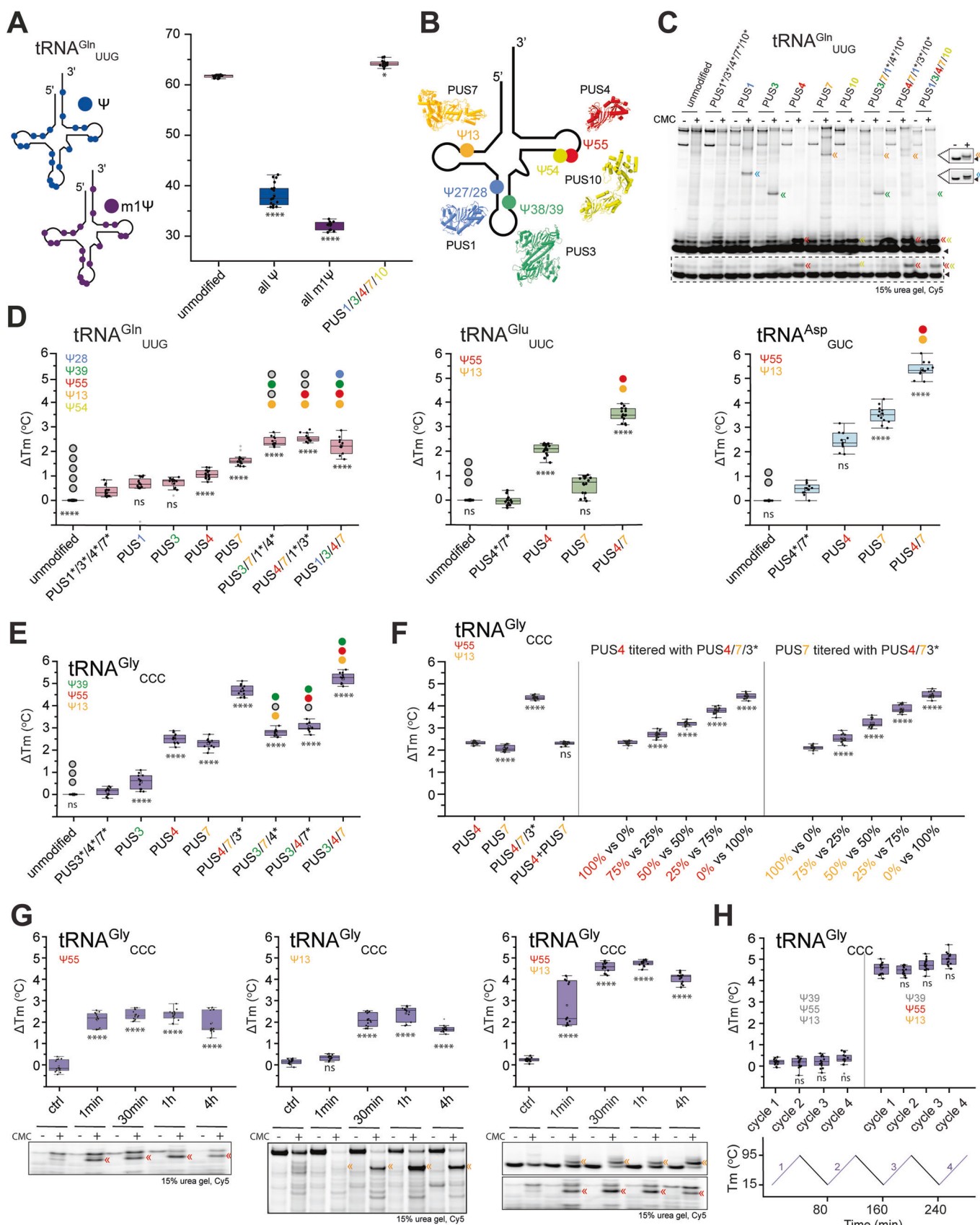

◄ **Figure 2. Effects of Ψ incorporation into in vitro modified human tRNAs.**

(A) Cloverleaf representations of a human tRNA$^{Gln}_{UUG}$ with all uridines (U) replaced by Ψ (blue) or m$^1$Ψ (violet). Melting temperatures (°C) of tRNA$^{Gln}_{UUG}$ with various modifications (Tm is indicated). Biological replicates, $N = 3$. (B) 2D representation of the structure of tRNA with site-specific Ψ (circles) added by individual PUS. PUS are shown in cartoon representation: PUS1, PUS3, and PUS4 are obtained from Alphafold2 prediction while PUS7 (PDB 5KKP) and PUS10 (PDB 2V9K) are crystal structures. (C) Detection of PUS-dependent Ψ formation on tRNA$^{Gln}_{UUG}$. The reverse-transcribed cDNA products were resolved in a 15% urea gel and the CMC-Ψ mediated short cDNAs product are indicated by double-arrows (Ψ$_{13}$ orange, Ψ$_{28}$ blue, Ψ$_{39}$ green, Ψ$_{55}$ red) while the Cy5-primer is indicated by a triangle. A short exposure of the same gel is shown in the dashed box. The Ψ$_{13}$ and Ψ$_{28}$-corresponding cDNAs are detected using a different set of site-specific primers (shown in a dash lined box on the right). (D) Melting temperature changes (ΔTm, °C) of tRNA$^{Gln}_{UUG}$, tRNA$^{Glu}_{UUC}$, tRNA$^{Asp}_{GUC}$ by Ψ modifications at specific positions in comparison to corresponding tRNA treated with a mixture of inactive PUS. Gray dots on the plot indicate unmodified tRNA and colored dots indicate introduced Ψ sites. Biological replicates, $N = 3$. (E) Melting temperature changes (ΔTm, °C) of tRNA$^{Gly}_{CCC}$ by Ψ modifications at specific positions in comparison to corresponding tRNA treated with a mixture of inactive PUS. Gray dots on the plot indicate unmodified tRNA and colored dots indicate introduced Ψ sites. Biological replicates, $N = 3$. (F) Melting temperature changes (ΔTm, °C) of tRNA$^{Gly}_{CCC}$ modified only by PUS4, only by PUS7, by PUS4 and PUS7 and after mixing the individual PUS4 and PUS7 samples at a 1:1 ratio (left). Melting temperature changes (ΔTm, °C) of tRNA$^{Gly}_{CCC}$ after mixing samples modified individually by PUS4 or PUS4/7 at different ratios (middle). Melting temperature changes (ΔTm, °C) of tRNA$^{Gly}_{CCC}$ after mixing samples modified individually by PUS7 or PUS4/7 at different ratios (middle). Biological replicates, $N = 3$. (G) Melting temperature changes (ΔTm, °C) of tRNA$^{Gly}_{CCC}$, after incubation with PUS4 (left) PUS7 (middle) and PUS4/7 (right) for the indicated time in comparison to corresponding tRNA not incubated with any PUS. Detection of PUS-dependent Ψ formation on tRNA$^{Gly}_{CCC}$ (below). Biological replicates, $N = 3$. (H) Melting temperature changes (ΔTm, °C) of unmodified (left) and PUS3, PUS4, and PUS7 modified (right) tRNA$^{Gly}_{CCC}$, after repeated heating and cooling cycles. Biological replicates, $N = 3$. The distribution of the data is represented in box plots. The central line inside the box indicates the median and the square the mean. The lower and upper edges correspond to the first (Q1) and third (Q3) quartiles, respectively, defining the interquartile range (IQR). The whiskers extend to the minimal and maximal values within the 1.5 IQR. Points beyond this range are considered outliers and are shown as transparent points. All statistical analyses were performed using one-way ANOVA ($a = 0.05$) with a Bonferroni multiple comparisons test. Statistically significant differences are indicated (ns: no significance, ****$p ≤ 0.0001$). Inactive PUS variants are indicated by *. Source data are available online for this figure.

repurified the tRNA after the modification reaction and subjected it to a temperature gradient. In stark contrast to the complete Ψ-substitution, we observed an increased thermostability (Tm +3.4 °C) for tRNA$^{Gln}_{UUG}$ carrying Ψ$_{13}$, Ψ$_{28}$, Ψ$_{39}$, Ψ$_{54}$, and Ψ$_{55}$ (Fig. 2A).

Next, we performed in vitro pseudouridylation reactions using individual PUS enzymes or various combinations thereof (Fig. 2C). We found that the activities of PUS1 and PUS3 on their own contribute very little to the overall increase in stability, whereas TRUB1/PUS4, PUS7, and PUS10-dependent Ψs are principally responsible for the elevated thermostability of modified tRNA$^{Gln}_{UUG}$. The Tm value is further increased when Ψ$_{13}$ (PUS7) is combined with Ψ$_{39}$ (PUS3; Tm +2.4 °C) or Ψ$_{55}$ (TRUB1/PUS4; Tm +2.6 °C), respectively. Incubation with a mixture of all inactive PUS enzymes shows no detectable Ψ modifications and no impact on the thermostability of tRNA$^{Gln}_{UUG}$ (Fig. 2D), supporting the specificity of the stabilizing effect by the Ψ modification itself. Of note, we noticed that PUS10 can catalyze pseudouridinylation of both, U$_{54}$ and U$_{55}$. Pseudouridinylation at position 55 by PUS4 does not seem to affect the activity of PUS10 in vitro or vice versa (Fig. EV4C). Therefore, it is difficult to analyze the impact of Ψ$_{54}$ on tRNA stability individually and we focused on the analyses on Ψ$_{13}$, Ψ$_{28}$, Ψ$_{39}$, and Ψ$_{55}$ for all tested human tRNAs.

We tested whether tRNA$^{Glu}_{UUC}$, tRNA$^{Asp}_{GUC}$ and tRNA$^{Gly}_{CCC}$ are also stabilized by post-transcriptional pseudouridinylation. Those human tRNAs naturally carry fewer Ψ-sites, because they have other nucleotides (e.g., A, C, G) or uridine modifications (e.g., m$^5$U) in the respective position. Our experiments with tRNA$^{Glu}_{UUC}$ (naturally modified at Ψ$_{13}$, Ψ$_{54}$, and Ψ$_{55}$) and tRNA$^{Asp}_{GUC}$ (naturally modified at Ψ$_{13}$ and Ψ$_{55}$) reveal similar stabilization patterns. The introduction of Ψ$_{13}$ and Ψ$_{55}$ in tRNA$^{Glu}_{UUC}$ is most beneficial whereas the addition of Ψ$_{54}$, which is rare in eukaryotes (Roovers et al, 2021), does not contribute to the stability of tRNA$^{Glu}_{UUC}$ by itself and even lowers the Tm value. tRNA$^{Asp}_{GUC}$ only carries Ψ$_{13}$ and Ψ$_{55}$, which both improve the stability and again show a synergistic effect in the double-modified tRNA. We introduced Ψ$_{13}$, Ψ$_{39}$, and Ψ$_{55}$—the only three modifications to

naturally occur in tRNA$^{Gly}_{CCC}$—which are located on each of its three domains (D-arm, ASL, and T-arm, respectively) (Fig. EV4D). The presence of all three Ψs in tRNA$^{Gly}_{CCC}$ increases its Tm value by 5.2 °C (Fig. 2E). The presence of Ψ$_{13}$ or Ψ$_{55}$ influenced the Tm value more than the presence of Ψ$_{39}$ and combining Ψ$_{13}$ with Ψ$_{55}$ is as efficient as modifying all three sites. We confirmed that only the simultaneous modification of Ψ$_{13}$ with Ψ$_{55}$ on the same tRNA molecule leads to the observed synergistic effect, as mixing tRNAs carrying either Ψ$_{13}$ or Ψ$_{55}$ at an equimolar ratio, did not lead to an increase in Tm (Fig. 2F). Generating heterogenous mixtures of modified samples by titration leads to intermediate effects on the Tm. Time course experiments confirm the almost complete level of Ψ incorporation (Fig. 2G), which is in line with the known fast reaction kinetics of PUS enzymes in vitro (Purchal et al, 2022). Finally, we show that Ψs are highly resistant to heat and that the Tm changes can be observed even after several cycles of folding and unfolding (Fig. 2H). Our analyses suggest that incorporation of Ψ at positions 13 and 55 in the D- and T-arm, respectively, consistently increases the stability of in vitro transcribed human tRNAs. The incorporation of Ψ in alternate sites, namely Ψ$_{28}$ and Ψ$_{39}$, seems to be mostly dispensable for the stability of these four human tRNAs.

## Single particle cryo-EM structures of modified human tRNAs

Our ability to resolve individual tRNA structures by cryo-EM for the first time gives us the opportunity to dissect the structural mechanism by which pseudouridinylation influences and stabilizes tRNAs. To this end, we determined the structures of human tRNA$^{Gln}_{UUG}$, tRNA$^{Gly}_{CCC}$, tRNA$^{Glu}_{UUC}$ and tRNA$^{Asp}_{GUC}$ before and after introducing Ψ$_{13}$ and Ψ$_{55}$ by single particle cryo-EM (Fig. 3A–D; Appendix Figs. S2–9). We obtained improved maps after modification, which enabled us to place consistent ensembles of models for all four human tRNAs with Q-scores expected at these resolutions (Pintilie et al, 2020) and small RMSD differences among the top-scoring models for each reconstruction (Figs. 3A–D and EV3A). Comparing modified tRNA$^{Gln}_{UUG}$ and tRNA$^{Gly}_{CCC}$ to

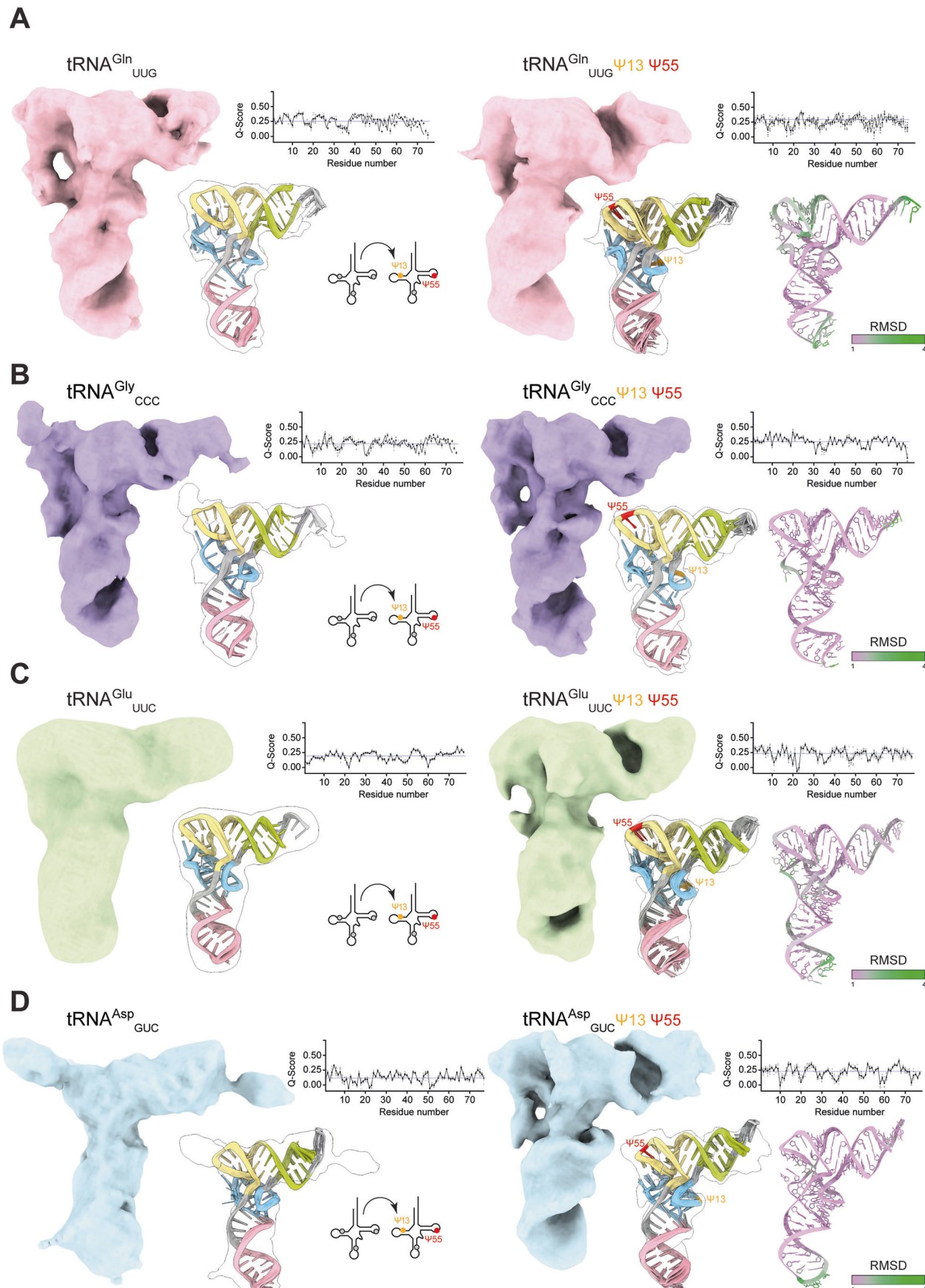

◀ **Figure 3.** **Analyses of in vitro modified human tRNAs.**

(A–D) Cryo-EM reconstructions of unmodified and $\Psi_{13}$- and $\Psi_{55}$-modified tRNA$^{Gln}_{UUG}$ (A), tRNA$^{Gly}_{CCC}$ (B), tRNA$^{Glu}_{UUC}$ (C), tRNA$^{Asp}_{GUC}$ (D). Cryo-EM reconstructions of unmodified and $\Psi_{13}$- and $\Psi_{55}$-modified human tRNA$^{Gln}_{UUG}$, tRNA$^{Gly}_{CCC}$, tRNA$^{Glu}_{UUC}$ and tRNA$^{Asp}_{GUC}$ with an ensemble of 10 atomic models fitted into the density. Modification sites marked: $\Psi_{13}$ in orange and $\Psi_{55}$ in red. Average Q-scores of the individual nucleotides are plotted for each structure. RMSD between unmodified and modified RMSD-colored difference between centroids of the ensembles of 10 atomic models of unmodified and modified tRNA fitted into the density. The modified sites are highlighted by color code in the models. All cryo-EM maps are contoured to RMSD = 8.

their unmodified counterparts, we observed changes in the elbow region and more distinct densities for the phosphate groups along the RNA backbone of the ASL and the AAS, indicating reduced disorder in the presence of $\Psi_{13}$ and $\Psi_{55}$. In the case of tRNA$^{Glu}_{UUC}$ and tRNA$^{Asp}_{GUC}$, the increase in rigidity provided by $\Psi_{13}$ and $\Psi_{55}$ modification led to an even more dramatic improvement of the cryo-EM densities: while the unmodified tRNAs produced feature-less maps of L-shaped blobs at intermediate resolution above 8 Å, stabilization of the D- and T-arms allowed us to resolve the individual arms of both tRNAs. Moreover, we observed a featureless map towards recognizable tRNA features at the elbow region of tRNA$^{Asp}_{GUC}$ upon modification (Fig. 3D). Of note, none of the observed conformational changes leads to a visible difference in migration behavior during native PAGE analyses (Fig. EV3B).

We also attempted to generate cryo-EM structures of these tRNAs with different patterns of pseudouridinylation. We obtained an interpretable map of tRNA$^{Gln}_{UUG}$ after simultaneously introducing $\Psi_{13}$ and $\Psi_{39}$. In this map, no compaction of the elbow region is observed, highlighting the importance of $\Psi_{55}$ for the flipping of the nucleotide and formation of the elbow (Fig. EV4E). In summary, we conclude that pseudouridinylation, particularly at the $\Psi_{13}$ and $\Psi_{55}$ sites, improves the interaction between D- and T-arms. Pseudouridinylation in other tRNA positions ($\Psi_{27/28}$ and $\Psi_{38/39}$) only marginally confers additional stability to tRNAs, showing that $\Psi$ acts highly position- and context-specific.

## Different $\Psi$ modification patterns have distinct effects on human tRNAs

Having established that $\Psi_{13}$ and $\Psi_{55}$ are key determinants of tRNA stability, we next asked whether these findings would hold true in other tRNA species that lack one or more commonly modified uridines or do not typically carry these canonical modifications. To address this, we extended our biophysical analyses to human tRNA$^{His}_{GUG}$, tRNA$^{Lys}_{UUU}$, tRNA$^{Ser}_{UGA}$ and tRNA$^{Arg}_{UCU}$ (Fig. 4A; Appendix Fig. S10), all of which display alternative $\Psi$ patterns in cells. We found that tRNA$^{His}_{GUG}$, which is naturally modified by PUS1 ($\Psi_{28}$), TRUB1/PUS4 ($\Psi_{55}$), and PUS7 ($\Psi_{13}$), is stabilized by $\Psi_{13}$ and $\Psi_{55}$, but in this case $\Psi_{28}$ does not contribute to additional stability. By contrast, tRNA$^{Lys}_{UUU}$, which contains three natural modification sites ($\Psi_{27}$, $\Psi_{39}$, and $\Psi_{55}$), is strongly stabilized by the presence of $\Psi_{27}$ and $\Psi_{39}$, while $\Psi_{55}$ seem to play a less prominent role. We speculate that the uridine-rich ASL loop may require additional stabilization with $\Psi_{27}$ and $\Psi_{39}$, two modified residues located within the ASL. tRNA$^{Ser}_{UGA}$ lacks U$_{13}$ and exhibits only a moderate stabilization following the introduction of $\Psi_{55}$, suggesting that its longer variable region makes this tRNA less dependent on the presence of D- and T-arm modifications. Instead, the stability of tRNA$^{Ser}_{UGA}$ seems to be dominated by the stabilization of the ASL by $\Psi_{27/28}$ and $\Psi_{39}$. Finally, tRNA$^{Arg}_{UCU}$ shows only a minimal

change in Tm (+1.2 °C) after introducing $\Psi$ at positions 28, 39, or 55. Hence, it seems that the stability of tRNA$^{Arg}_{UCU}$ is not dependent on the presence of $\Psi$, which raises the question of whether the modifications could be beneficial for this tRNA in some other way.

## Local structural stabilization of tRNAs by $\Psi$

As the conversion from U to $\Psi$ would not lead to altered features of the coulomb potential map of the nucleobase, we would not expect to visualize the modifications in our structural data even at high resolution. Nonetheless, we reasoned that we may be able to detect the resultant local structural consequences of pseudouridinylation in our cryo-EM maps. We calculated and compared local resolution maps of tRNA$^{Gln}_{UUG}$ and tRNA$^{Gly}_{CCC}$ before and after introducing different sets of modifications, namely $\Psi_{13}/\Psi_{39}$ or $\Psi_{13}/\Psi_{55}$ and $\Psi_{13}/\Psi_{55}$ or $\Psi_{13}/\Psi_{39}/\Psi_{55}$, respectively. As explored above, we find that pseudouridinylation impacts on local conformation and improves the local resolution in regions around the respective modification sites to a greater extent than more distal regions of the same tRNA (Fig. 4B; Appendix Figs. S11 and S12; Movie EV3 and EV4). More specifically, the presence of $\Psi_{39}$ reduces the disorder in the ASL of tRNA$^{Gln}_{UUG}$, whereas $\Psi_{55}$ induces the compaction of the elbow region in both analyzed tRNAs. For instance, in tRNA$^{Gln}_{UUG}$, the elbow region seems to get loosened when U$_{13}$ and U$_{39}$ are modified to $\Psi$. Upon U$_{55}$ modification, the resultant region gets compact again (like in the unmodified state), highlighting an example where $\Psi$ provides a local structural switch on the modified residue itself. The conditional nature of U$_{55}$ accessibility shows that certain modifications may only be possible or functional in the presence of others (Sokolowski et al, 2018).

To gain further insight of how the incorporation of $\Psi$ can lead to stabilization, we carried out molecular dynamics simulations of folded tRNA$^{Gln}_{UUG}$ carrying no modifications, $\Psi_{13}$, $\Psi_{55}$ or $\Psi_{13}/\Psi_{55}$ (500 ns, three independent simulations for each of the four variants). We analyzed the interactions of all tRNA nucleotides during the simulation with a specific focus on the respective Us and $\Psi$s and their close surrounding. In all modified variants, $\Psi_{13}$ and $\Psi_{55}$ residues formed additional water-mediated hydrogen bonds between the N1-H group and the phosphate oxygen of the preceding 5′ residue, and $\Psi_{13}$ additionally tended to form an additional hydrogen bond to its own phosphate. These interactions, missing for the U residues, lowered the mean $\Psi_{13}$ and $\Psi_{55}$ stacking energies by ~4.1 and ~1.8 kcal/mol, respectively, thereby improving tRNA stability (Fig. EV5).

To complement our structural analyses, we measured the Temperature-related Intensity Change (TrIC) of tRNA variants carrying site-specific labels over an increasing temperature gradient (Fig. 4C). For these experiments, we first synthesized full-length tRNA$^{Gln}_{UUG}$ with a Cy5-probe positioned either at the

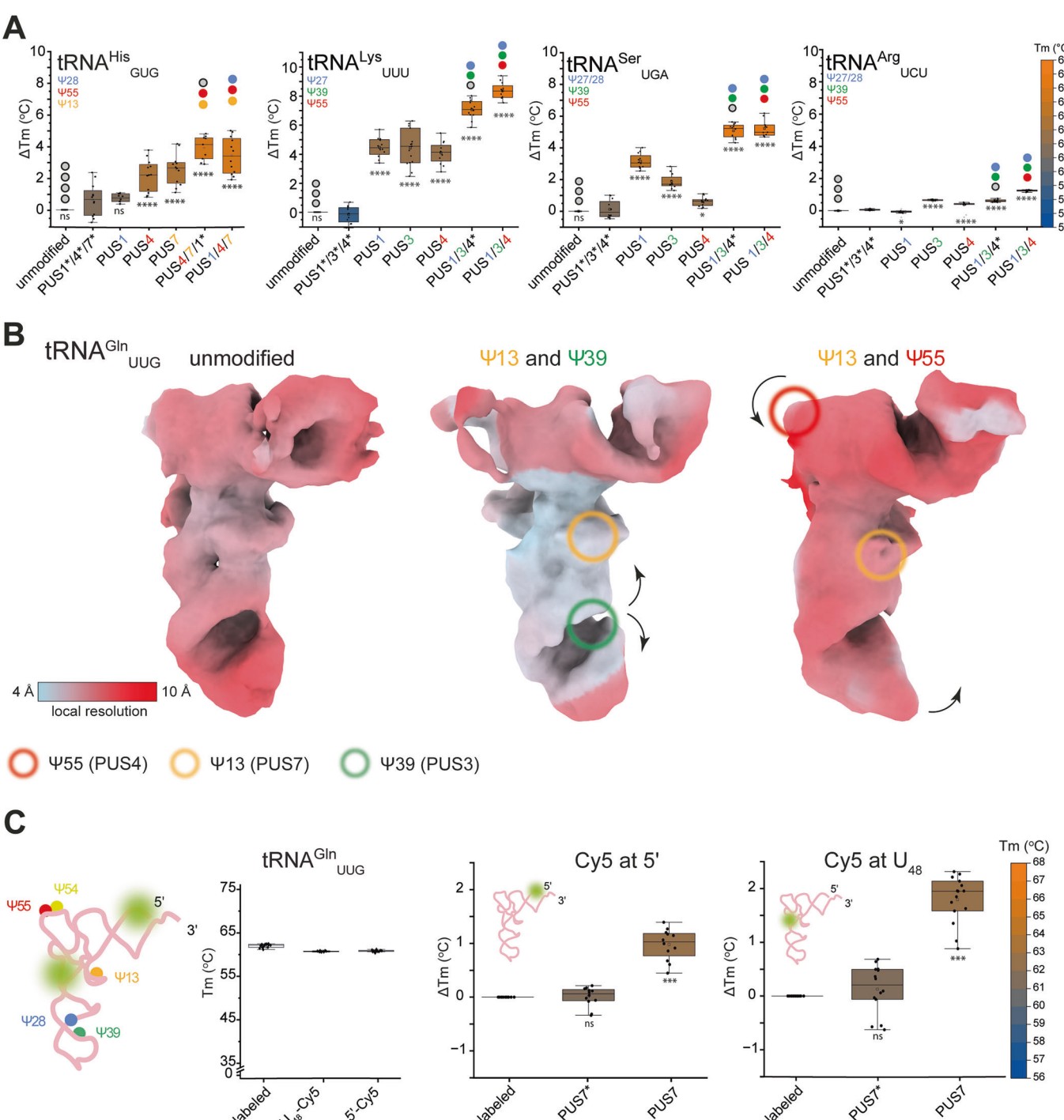

**Figure caption region:**

Ψ55 (PUS4)   Ψ13 (PUS7)   Ψ39 (PUS3)

5′-end or close to the interface between D- and T-arm ($U_{48}^{Cy5}$). The Tm value of the 5′-labeled tRNA (without Ψ) agrees well with our earlier measurements using RiboGreen™ (Fig. 2D). We then measured Tm values of the two fluorescently labeled versions of tRNA$^{Gln}_{UUG}$ treated with either PUS7 to install $\Psi_{13}$ [which caused the largest Tm-increase when applied individually (Fig. 2D)], or with the mixture of five PUS enzymes to catalyze the introduction of all naturally-occurring Ψ sites. We observed that both 5′- and $U_{48}$-labeled tRNAs display a Ψ-induced thermostability change.

The relative Tm shifts after Ψ incorporation are almost identical for RiboGreen™ and the 5′-probe, whereas the relative Tm difference for the $U_{48}^{Cy5}$-labeled tRNA$^{Gln}$ is significantly higher (Fig. 4C). Therefore, $\Psi_{13}$ indeed stabilizes the proximal nucleotides around the core to a higher extent than the residues in other elements of the tRNA. In summary, Ψ enhances tRNA structural rigidity locally, resulting in a site-specific stabilization effect that depends on its surrounding structural context (Voegele et al, 2023).

**Figure 4.   Ψ incorporation has different effects on tRNA local stability and structure.**

(A) Melting temperature changes ($\Delta$Tm, °C) of tRNA$^{His}_{GUG}$, tRNA$^{Lys}_{UUU}$, tRNA$^{Ser}_{UGA}$, tRNA$^{Arg}_{UCU}$ by Ψ modifications at specific positions in comparison to corresponding tRNA treated with a mixture of inactive PUS. Gray dots on the plot indicate unmodified tRNA and colored dots indicate introduced Ψ sites. Biological replicates, $N = 3$. (B) Local resolution estimations for tRNA$^{Gln}_{UUG}$. The relative resolution scale is shown in the inset. Ψ sites are highlighted by circles (Ψ$_{13}$ orange, Ψ$_{39}$ green, Ψ$_{55}$ red). The arrows indicate the local conformational changes. All cryo-EM maps are contoured to RMSD = 8. (C) tRNA representation with Ψ sites and the positions of Cy5 probe are highlighted (green). Global melting temperatures (Tm, °C) measured with RiboGreen for unlabeled and U$_{48}$ or 5′-Cy5 labeled tRNA$^{Gln}_{UUG}$. Melting temperature changes ($\Delta$Tm, °C) of tRNA$^{Gln}_{UUG}$ labeled with Cy5 at 5′ (left) or U$_{48}$ (right) upon Ψ modifications. The temperature (Tm) scale is shown in the inset. Biological replicates, $N = 3$. Inactive PUS are indicated by *. The distribution of the data is represented in box plots. The central line inside the box indicates the median and the square the mean. The lower and upper edges correspond to the first (Q1) and third (Q3) quartiles, respectively, defining the interquartile range (IQR). The whiskers extend to the minimal and maximal values within the 1.5 IQR. Points beyond this range are considered outliers and are shown as transparent points. All statistical analysis was performed using one-way ANOVA ($\alpha = 0.05$) with a Bonferroni multiple comparisons test. Statistically significant differences are indicated (ns: no significance, ****$p \leq 0.0001$. For tRNA$^{Ser}_{UGA}$ *$p = 0.01517$, for tRNA$^{Arg}_{UCU}$ *$p = 0.04746$). Source data are available online for this figure.

## Cryo-EM structures of endogenous human tRNAs

Our work up until this point has relied on in vitro transcribed tRNAs that provide an ideal pure and controllable substrate, but we next asked whether these methodologies could extend to endogenous tRNAs as a more biologically relevant system. However, it is hard to isolate homogeneous samples of a single tRNA iso-decoder because of the similar shape, sequence and charge of the cellular tRNA pool. We isolated thiolated human tRNA$^{Arg}_{UCU}$ and tRNA$^{Lys}_{UUU}$ from HEK293T cells using a modified chaplet column chromatography (CCC) method (Suzuki and Suzuki, 2007) and analyzed their structures by cryo-EM. We used thiolated tRNAs to take advantage of APM gels (Fig. 5A) that allowed us to confirm the high purity of our CCC preparations. Both samples yielded maps resembling the canonical tRNA size and shape that could be used to fit sequence-based models. The obtainable resolution (~10 Å) for the endogenous tRNAs was limited in comparison to the well-defined maps of the in vitro-derived tRNAs and the reasons could be manifold. Nonetheless, the modification patterns might also be heterogenous for the same iso-decoder tRNA, which can hamper high-resolution reconstructions. The incorporation of additional, larger modifications (e.g., mcm$^5$s$^2$U$_{34}$, t$^6$A$_{37}$), might decrease the stability of a given tRNA again, leading to an increased entropy and flexibility of fully modified tRNAs (Fig. 5B).

Future studies will need to focus on solving these issues by optimizing purification protocols for each individual tRNA, by customizing sample preparation methods, by further improving cryo-EM data collection strategies and data analyses pipelines. In addition, our study focuses exclusively on Ψs and the used in vitro transcribed tRNAs lack other modifications. Hence, future studies need to include the full range of tRNA modification to understand how they contribute to folding and stability of tRNAs and how they affect the stabilizing effects of Ψs, determined in the study. Nonetheless, our proof-of-concept data paves the way toward structural characterization of endogenous tRNAs and other similar-sized structured RNA molecules from endogenous sources.

## Discussion

Our biophysical measurements of 12 human tRNAs and structural analyses of 6 human tRNAs, carrying unique sets of modifications and originating from in vitro transcribed- or endogenous sources, reveal fundamental principles of tRNA biology that address many longstanding questions in the field. The wide range of different tRNAs we analyzed allowed us to show that tRNA sequences differ in their capability to fold into a functional tRNA shape on their own and define a crucial role for stable interaction between the T- and D-arms to facilitate folding (Fig. 5C). By introducing defined sets of Ψ modifications, we demonstrate that the presence of Ψ$_{13}$ and Ψ$_{55}$ increases the thermostability of almost all tested tRNAs, whereas Ψ$_{27/28}$ and Ψ$_{39}$ are mostly dispensable for thermostability, but may confer a small stabilizing effect in individual tRNA molecules. We would like to mention that due to the limited resolution of our cryo-EM reconstructions, we are not able to directly observe the conformation of the introduced Ψs. Using molecular dynamics simulations, we provide a mechanistic explanation for the observed stabilization, which we inferred indirectly from local resolution improvements. Hence, our work shows that Ψ act locally to stabilize certain tRNA subdomains, which we reason may help compensate for suboptimal sequences in those regions.

Our work also conclusively shows that all five human stand-alone PUS synthases do not depend on any priming activity of each other or of other RNA modification enzymes to function in vitro and do not act in a mutually exclusive manner. Furthermore, four of the five purified human PUS enzymes show site- and tRNA-specificity in vitro without the presence of other tRNA modifications. The specificities and recognition mechanisms of PUS1 (Carlile et al, 2019), PUS3 (Lin et al, 2024), TRUB1/PUS4 (Carlile et al, 2014), PUS7 (Guegueniat et al, 2021) and PUS10 (Deogharia et al, 2019) have been studied. The necessity of site-specific pseudouridinylation is in line with the co-evolutionary connection between the appearance of tRNA modifying enzymes and the number of sequences that can fold into a functional tRNA (Novoa et al, 2012; McKenney et al, 2017). Our observations for human PUS10 recapitulate findings in a previous report (Gurha and Gupta, 2008), describing the specificity of archaeal PUS10 for both neighboring positions, namely U54 and U55. Additional work will be necessary to fully understand the molecular mechanisms that lead to the specificity of human PUS10 in vivo (Deogharia et al, 2019) and to reveal the functional consequences of the reduced stability after introducing Ψ$_{54}$ in combination with other Ψ (i.e., Ψ$_{55}$). We show that enzymatically inactive stand-alone PUS proteins, which still tightly bind tRNAs, do not increase the thermostability of tRNAs—providing strong evidence that the increased tRNA stability is a consequence of the modifications themselves. Finally, we show that replacing all uridines with Ψ-derivatives during in vitro transcription has a detrimental effect on the folding of tRNAs. Since current mRNA vaccines are completely

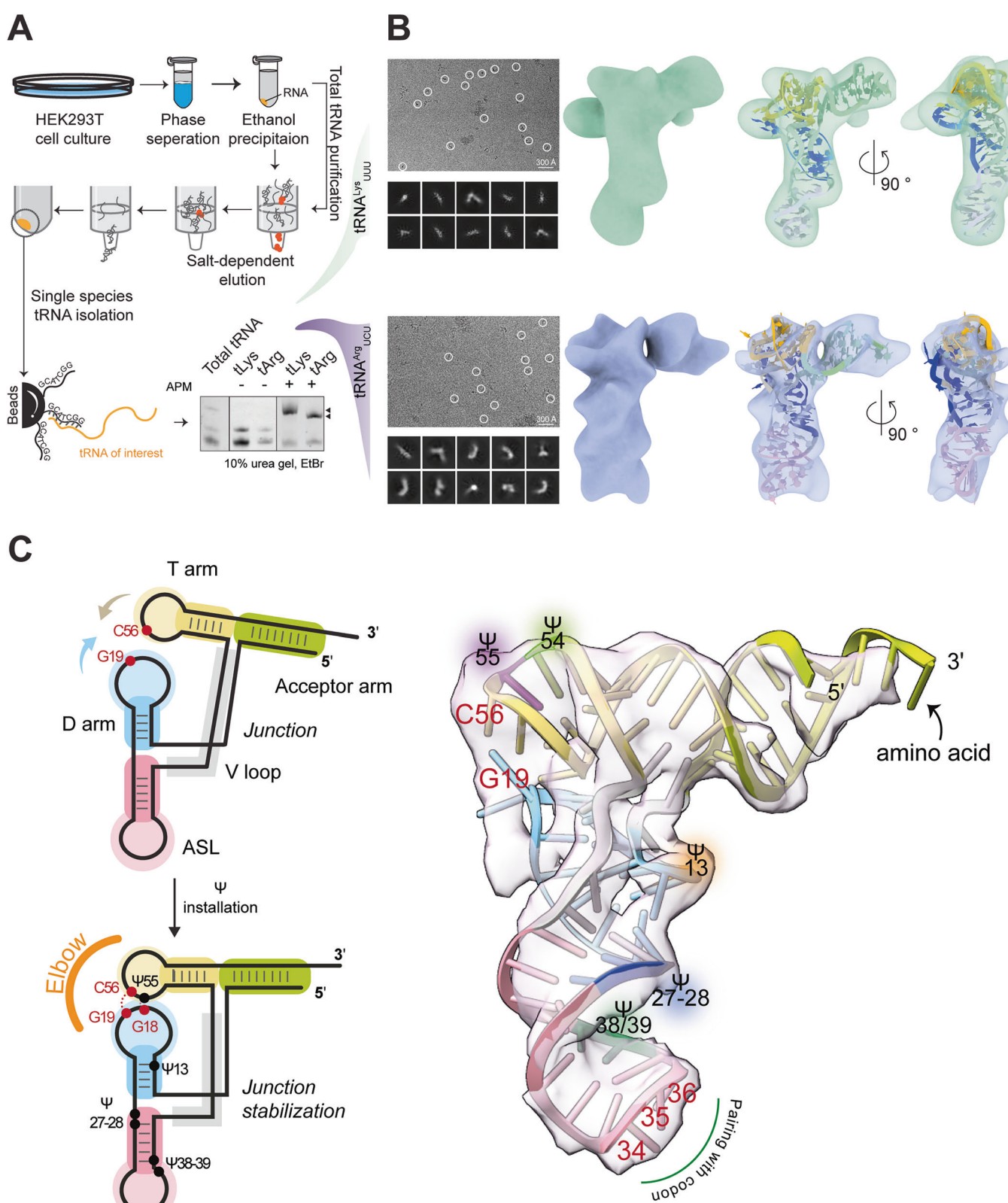

modified by Ψ, these findings have far-reaching consequences for the potential use of folded RNA domains in next-generation RNA drugs. Our work certainly highlights that the specific incorporation of Ψ could be very beneficial for the stability of RNA domains, while saturating substitution could have unfavorable consequences. Moreover, the effect of Ψ is indeed strongly dependent on context, which coincides with a recent observation by NMR in small RNA motifs (Voegele et al, 2023). Our findings demonstrate that Tm

**Figure 5. Structural analyses of endogenous tRNAs from human cells.**

(A) Scheme of isolation of cytosolic tRNA from HEK293T cells. Isolated tRNA$^{Lys}_{UUU}$ and tRNA$^{Arg}_{UCU}$ were resolved in a 10% urea gel with and without addition of APM. The tRNA$^{Lys}_{UUU}$ and tRNA$^{Arg}_{UCU}$ migrate slower in the APM gel (indicated by a triangle). (B) Representative micrograph and 2D class averages of human tRNA$^{Lys}_{UUU}$ (upper left) and tRNA$^{Arg}_{UCU}$ (lower left). Cryo-EM reconstructions of human tRNA$^{Lys}_{UUU}$ (upper right) and tRNA$^{Arg}_{UCU}$ (lower right) with a model built into the density (right). All cryo-EM maps are contoured to RMSD = 8. (C) Left: Secondary tRNA structure representations explaining how Ψ incorporation influences tRNA tertiary structure, mainly by elbow and junction regions stabilization. Right: tRNA$^{Gly}_{CCC}$ density with the model built in and indicated all possible Ψ modification sites. Source data are available online for this figure.

values report on global as well as local stabilizing effects caused by introducing Ψ into tRNAs. However, this relatively coarse measurement that does not capture finer aspects of folding kinetics, local flexibility, or intermediate states. Therefore, this measurement needs to be combined with complementary approaches (e.g., FRET, NMR, AFM) to further enhance our understanding of tRNA folding and dynamics.

Structural information on isolated human tRNAs has been limited up until this point, with only one single crystal structure of a human tRNA$^{Sec}$ available (PDB 3A3A) (Itoh et al, 2009). While most of the previous studies on tRNA processing, modifying, and editing enzymes were restricted to the same few highly abundant and stable tRNAs (e.g., tRNA$^{Phe}$), here we provide a relatively simple technical solution to characterize the structure and function of all tRNA families in the future. Ultimately, the use of PUS enzymes to modify and stabilize tRNAs produced by IVT will vastly increase the sequence space of tRNAs available for analysis from any species, enabling better structural biology approaches and facilitating the identification of substrate specificities in vitro. As Ψ is present in a variety of RNA families (Begik et al, 2021), our approach of using Ψ to aid tRNA structure determination will inspire a new way to characterize other RNA domains, including introns (Martinez et al, 2022) and long non-coding RNAs.

Our cryo-EM work on individual tRNAs—representing one of the smallest asymmetric RNA particles analyzed by cryo-EM to date (Kappel et al, 2020; Kretsch et al, 2024; Chen et al, 2024)—

paves the way toward directly studying the consequences of tRNA folding and processing (e.g., tRNA splicing, modification, aminoacylation) before and after the reaction. Beyond the structural studies we performed, the combination of simulations and biophysical analyses (e.g., DLS, DSF and TrIC) could be applied to other structured RNAs and RNA domains to facilitate the identification of long-range interactions, stabilizing RNA modifications, fold-defining elements and samples suitable for structural analyses. Our work suggests that it will be possible to use a combination of cryo-EM, biophysical experiments and RNA folding/unfolding simulations to understand pre-tRNAs (Randau and Söll, 2008), armless tRNAs present in the mitochondria of certain species (Jühling et al, 2018), suppressor tRNAs (Kachale et al, 2023), ACEtRNAs (Lueck et al, 2019), 4-codon tRNAs (DeBenedictis et al, 2021) and other biotechnologically relevant tRNAs in the future. In addition, it remains to be shown, whether Ψ plays a similar role for the folding and stability of these atypical tRNAs. We are particularly excited to analyze rare, clinically relevant mutations in tRNA sequences (Lai et al, 2022) to understand their structural consequences in future work. Ultimately, our work highlights the vast potential for understanding the structure and function of small and highly dynamic RNA molecules by cryo-EM (Liu et al, 2022; Ma et al, 2022).

## Methods

**Reagents and tools table**

| Reagent/Resource | Reference or Source | Identifier or Catalog Number |
|---|---|---|
| **Recombinant DNA** | | |
| pETM30 expression vector | Addgene | Plasmid #133426 |
| pETM11 expression vector | Addgene | Plasmid #108943 |
| pFastBacHT A expression vector | Invitrogen | 10584027 |
| **Oligonucleotides and other sequence-based reagents** | | |
| GTCTCTGTGGCGCAATGGACGAGCGCGCTGGACT TCTAATCCAGAGGTTCCGGGTTCGAGTCCCGGCA GAGATG | In house in vitro transcribed human tRNA Arg_TCT_4-1 | |
| GGTCCTCGTTAGTATAGTGGTGAGTATCCCCGCCT GTCACGCGGGAGACCGGGGTTCGATTCCCCGACG GGGAGCCA | In house in vitro transcribed human tRNA Asp_GTC_2-1 | |
| GGCCCCATGGTGTAATGGTTAGCACTCTGGACTT TGAATCCAGCGATCCGAGTTCAAATCTCGGTGG GACCTCCA | In house in vitro transcribed human tRNA Gln_TTG_3-1 | |
| GGTCCCTGGTGGTCTAGTGGCTAGGATTCGGCGC TTTCACCGCCGCGGCCCGGGTTCGATTCCCGGTC AGGGAACCA | In house in vitro transcribed human tRNA Glu_TTC_4-1 | |

| Reagent/Resource | Reference or Source | Identifier or Catalog Number |
|---|---|---|
| GCGCCGCTGGTGTAGTGGTATCATGCAAGATTCCCATTCTTGCGACCCGGGTTCGATTCCCGGGCGGCGCACCA | In house in vitro transcribed human tRNA Gly_CCC_2-1 | |
| GCCGTGATCGTATAGTGGTTAGTACTCTGCGTTGTGGCCGCAGCAACCTCGGTTCGAATCCGAGTCACGGCACCA | In house in vitro transcribed human tRNA His_GTG_1-1 | |
| GCCCGGATAGCTCAGTCGGTAGAGCATCAGACTTTTAATCTGAGGGTCCAGGGTTCAAGTCCCTGTTCGGGCGCCA | In house in vitro transcribed human tRNA Lys_TTT_3-1 | |
| GGCTCGTTGGTCTAGGGGTATGATTCTCGCTTTGGGTGCGAGAGGTCCCGGGTTCAAATCCCGGACGAGCCCCCA | In house in vitro transcribed human tRNA Pro_TGG_3-1 | |
| GGCAGCGATGGCCGAGTGGTTAAGGCGTTGGACTTGAAATCCAATGGGGTCTCCCCGCGCAGGTTCGAACCCTGCTCGCTGCGCC | In house in vitro transcribed human tRNA Ser_TGA_1-1 | |
| GGTTTCCGTGGTGTAGTGGTTATCACATTCGCCTTACACGCGAAAGGTCCTCGGGTCGAAACCGAGCGGAAACACCA | In house in vitro transcribed human tRNA Val_TAC_4-1 | |
| GACTCTGGTGGGACTCGAACCCACAACCTT | IDT 3'biotinylated oligonucleotide for human tRNA Arg_UCU | Custom DNA synthesis |
| TGGCGCCCGAACAGGGACTTGAACCCTG | IDT 3'biotinylated oligonucleotide for human tRNA Lys_UUU | Custom DNA synthesis |
| TTCTAATACGACTCACTATAGTCTCTGTGGCGCAATGGACGAGCGCGCTGGACTT | Merck, primer for Arg_TCT_4-1_F | Custom DNA synthesis |
| CATCTCTGCCGGGACTCGAACCCGGAACCTCTGGATTAGAAGTCCAGCGCGCTC | Merck, primer for Arg_TCT_4-1_R | Custom DNA synthesis |
| TTCTAATACGACTCACTATATCCTCGTTAGTATAGTGGTGAGTATCCCCGCCTG | Merck, primer for Asp_GTC_2-1_F | Custom DNA synthesis |
| TTCTAATACGACTCACTATAGTCCTCGTTAGTATAGTGGTGAGTATCCCCGCCTG | Merck, primer for Asp_GTC_2-1_F_g | Custom DNA synthesis |
| TTCTAATACGACTCACTATAGGTCCTCGTTAGTATAGTGGTGAGTATCCCCGCCTG | Merck, primer for Asp_GTC_2-1_F_gg | Custom DNA synthesis |
| TGGCTCCCCGTCGGGGAATCGAACCCCGGTCTCCCGCGTGACAGGCGGGGATACTCACC | Merck, primer for Asp_GTC_2-1_R | Custom DNA synthesis |
| TTCTAATACGACTCACTATAGGCCCCATGGTGTAATGGTTAGCACTCTGGACTTTGAAT | Merck, primer for Gln_TTG_3-1_F | Custom DNA synthesis |
| TGGAGGTCCCACCGAGATTTGAACTCGGATCGCTGGATTCAAAGTCCAGAGTGCTAAC | Merck, primer for Gln_TTG_3-1_R | Custom DNA synthesis |
| TTCTAATACGACTCACTATATCCCTGGTGGTCTAGTGGCTAGGATTCGGCGCTT | Merck, primer for Glu_TTC_4-1_F | Custom DNA synthesis |
| TTCTAATACGACTCACTATAGTCCCTGGTGGTCTAGTGGCTAGGATTCGGCGCTT | Merck, primer for Glu_TTC_4-1_F_g | Custom DNA synthesis |
| TTCTAATACGACTCACTATAGGTCCCTGGTGGTCTAGTGGCTAGGATTCGGCGCTT | Merck, primer for Glu_TTC_4-1_F_gg | Custom DNA synthesis |
| TGGTTCCCTGACCGGGAATCGAACCCGGGCCGCGGCGGTGAAAGCGCCGAATCCTAGC | Merck, primer for Glu_TTC_4-1_R | Custom DNA synthesis |
| TTCTAATACGACTCACTATAGCGCCGCTGGTGTAGTGGTATCATGCAAGATTCCCATT | Merck, primer for Gly_CCC_2-1_F | Custom DNA synthesis |
| TGGTGCGCCGCCCGGGAATCGAACCCGGGTCGCAAGAATGGGAATCTTGCATGATACC | Merck, primer for Gly_CCC_2-1_R | Custom DNA synthesis |
| TTCTAATACGACTCACTATAGCCGTGATCGTATAGTGGTTAGTACTCTGCGTTGTGGCCGCAGCAACCT | Merck, primer for His_GTG_1-1_F | Custom DNA synthesis |
| TGGTGCCGTGACTCGGATTCGAACCGAGGTTGCTGCGGCCACA | Merck, primer for His_GTG_1-1_R | Custom DNA synthesis |
| TTCTAATACGACTCACTATAGCCCGGATAGCTCAGTCGGTAGAGCATCAGACTTTTAATC | Merck, primer for Lys_TTT_3-1_F | Custom DNA synthesis |
| TGGCGCCCGAACAGGGACTTGAACCCTGGACCCTCAGATTAAAAGTCTGATGCTCTACCG | Merck, primer for Lys_TTT_3-1_R | Custom DNA synthesis |

| Reagent/Resource | Reference or Source | Identifier or Catalog Number |
|---|---|---|
| TTCTAATACGACTCACTATAGGCTCGTTGGTCTAGGGGTATGATTCTCGCTTTGGGTGC | Merck, primer for Pro_TGG_3-1_F | Custom DNA synthesis |
| TGGGGGCTCGTCCGGGATTTGAACCCGGGACCTCTCGCACCCAAAGCGAGAATCAT | Merck, primer for Pro_TGG_3-1_R | Custom DNA synthesis |
| TTCTAATACGACTCACTATAGCAGCGATGGCCGAGTGGTTAAGGCGTTGGACTTGAAA | Merck, primer for Ser_TGA_1-1_F | Custom DNA synthesis |
| TTCTAATACGACTCACTATAGGCAGCGATGGCCGAGTGGTTAAGGCGTTGGACTTGAAA | Merck, primer for Ser_TGA_1-1_F_g | Custom DNA synthesis |
| TGGCGCAGCGAGCAGGGTTCGAACCTGCGCGGGGAGACCCCATTGGATTTCAAGTCCAACGCCTTAAC | Merck, primer for Ser_TGA_1-1_R | Custom DNA synthesis |
| TTCTAATACGACTCACTATAGTTTCCGTGGTGTAGTGGTTATCACATTCGCCTTACACGC | Merck, primer for Val_TAC_4-1_F | Custom DNA synthesis |
| TTCTAATACGACTCACTATAGGTTTCCGTGGTGTAGTGGTTATCACATTCGCCTTACACGCGAAAGGTCCTCGGG | Merck, primer for Val_TAC_4-1_F_g | Custom DNA synthesis |
| TGGTGTTTCCGCTCGGTTTCGACCCGAGGACCTTTCGCGTGTAAGGCGAATGTGAT | Merck, primer for Val_TAC_4-1_R | Custom DNA synthesis |
| 5'-ggc ccc aug gug uaa ugg uua gca cuc ugg acu uug aau cca gcg a(Cy5-U)c cga guu caa auc ucg gug gga ccu cca-3' | Gln-TTG-3-1_U48Cy5 https://www.metabion.com/ | Custom RNA synthesis |
| 5'-Cy5-ggc ccc aug gug uaa ugg uua gca cuc ugg acu uug aau cca gcg auc cga guu caa auc ucg gug gga ccu cca-3' | Gln-TTG-3-1_5'-Cy5 https://www.metabion.com/ | Custom RNA synthesis |
| **Chemicals, Enzymes and other reagents** | | |
| RiboGreen | ThermoFisher Scientific | R11491 |
| Oil-based liquid rubber | NanoTemper Technologies | PR-P001 |
| Andromeda Capillaries | NanoTemper Technologies | AN-041001 |
| Prometheus High Sensitivity Capillaries | NanoTemper Technologies | PR-C006 |
| Acid Phenol Chloroform | ThermoFisher Scientific | AM9722 |
| Sodium chloride | Epro | SOD002.5 |
| Potassium Chloride | Linegal Chemicals | ROTH-HN02.3 |
| Magnesium Chloride hexahydrate | Linegal Chemicals | 2189.1 |
| Tris-HCl | Epro | 10104159001 |
| DTT | Epro | DTT 001.25 |
| Ethanol 96% 500 mL | Chemland | STLAB*0016-500 ML |
| Imidazole | Epro | IMD508.250 |
| Urea | Linegal Chemicals | ROTH-X999.1 |
| β -Mercaptoethanol | Epro | MER002.100 |
| DNase I | Merck | 10104159001 |
| HEPES | Epro | HEP001.1 |
| Ammonium chloride | Linegal Chemicals | 5470.4 |
| Ammonium acetate | Merck | 5438340100 |
| EDTA | LabEmpire | IB70180 |
| Benzonase | Merck | E1014-5KU |
| Roti®ZOL RNA, for molecular biology | Linegal Chemicals | 9319.2 |
| SuperScript™ III Reverse Transcriptase | ThermoFisher Scientific | 18080085 |
| N-Cyclohexyl-N'-(2-morpholinoethyl) carbodiimide methyl-p-toluenesulfonate (CMC) | Merck | C106402-1G |
| Pierce™ Protease Inhibitor Mini Tablets, EDTA-free | ThermoFisher Scientific | A32955 |
| Bicine | Merck | B3876-100G |

| Reagent/Resource | Reference or Source | Identifier or Catalog Number |
|---|---|---|
| Sodium bicarbonate, 1 M buffer soln., pH 10.0 | ThermoFisher Scientific | J63025.AK |
| Chloroform | Chempur | CHEM*112344305-1L |
| Triton X100 | LabEmpire | TRX777.100 |
| Sodium deoxycholate | Merck | D6750-10G |
| Sodium acetate | VWR | 0602-1KG |
| Glycogen | Merck | 10901393001 |
| BisTris HCl | VWR | 0715-100G |
| [N-acryloylamino] phenyl) mercuric chloride (APM) | Kind gift of Sebastian Leidel (University of Bern, Switzerland) | |
| IPTG | AA Biotechnology | 2003-25 |
| Glycerol | Chemland | 08CL00C0724.01000 |
| Protease K | Merck | P2308-10MG |
| dNTP mix | ThermoFisher Scientific | R0192 |
| ATP | ThermoFisher Scientific | R0441 |
| CTP | ThermoFisher Scientific | R0451 |
| GTP | ThermoFisher Scientific | R0461 |
| UTP | ThermoFisher Scientific | R0471 |
| Spermidine 0.1 M | Merck | 05292-1ML-F |
| 5-Propargylamino-CTP-Cy5 | Jena Bioscience | NU-831-CY5 |
| RNase-free DNase I | ThermoFisher Scientific | 89836 |
| RiboLock RNase Inhibitor | ThermoFisher Scientific | EO0382 |
| pyrophosphatase | ThermoFisher Scientific | EF0221 |
| HF buffer | ThermoFisher Scientific | F518L |
| Phusion polymerase | ThermoFisher Scientific | R0461 |
| T7 RNA polymerase | Produced in house | |
| DMEM HG | Lonza | BE12-604F |
| Trypsin-EDTA 0,25% | Corning Diag-Med | 25-053-CI |
| PBS w/o Mg++ & Ca++ | Lonza | BE17-516F |
| FBS | EurX | E5050-03 |
| BL21 (DE3) CodonPlus-RIL competent cells | ThermoFisher Scientific | C600003 |
| DH10Bac cells | ThermoFisher Scientific | 10361012 |
| Hi5 insect cells | ThermoFisher Scientific | B85502 |
| Sf9 (Expression Systems) insect cells | ThermoFisher Scientific | B82501 |
| FuGENE® HD Transfection Reagent | Promega | E2311 |
| **Software** | | |
| PR.Panta Control software v1.4.4 | NanoTemper Technologies | PR-S061 |
| AN.Control Software v1.1 | NanoTemper Technologies | AN-020001 |
| OriginPro | https://www.originlab.com/ | OriginPro 2023 10.0.0.154 |
| cryoSPARC | Structura Biotechnology Inc. | |
| SimRNA | https://genesilico.pl/software/stand-alone/simrna | v3.2 |
| **Other** | | |
| HEK293T cells | Dharmacon | |
| NucleoBond AX100 | Macherey-Nagel (VWR) | 740521 |
| Dynabeads MyOne Streptavidin C1 magnetic beads | ThermoFisher Scientific | 65002 |
| QUANTIFOIL® R 2/1 copper grids (200 mesh) | EM Resolutions Ltd. | QR21200Cu100 |

| Reagent/Resource | Reference or Source | Identifier or Catalog Number |
|---|---|---|
| NiNTA beads | Qiagen | 30230 |
| GSTPrep | Cytiva | 28936550 |
| Andromeda | NanoTemper Technologies | |
| Prometheus Panta | NanoTemper Technologies | |

## In vitro production of tRNAs

In vitro transcribed tRNAs were produced as previously described (Lin et al, 2022). PCR was used to generate the DNA templates with T7 polymerase recognition sequence at 5′ end. The tRNA sequences were retrieved from tRNA database (http://gtrnadb.ucsc.edu/) (Chan and Lowe, 2016) and the CCA-end was also introduced in the template sequences (Appendix Fig. S1A). To obtain sufficient material during T7-mediated IVT (Dunx et al, 1983), we introduced additional Gs at the 5′-end of tRNA$^{Asp}_{GUC}$, tRNA$^{Glu}_{UUC}$, tRNA$^{Ser}_{UGA}$, and tRNA$^{Val}_{UAC}$ (Appendix Fig. S1A). As it has been reported that the addition of 5′-nucleotides can impacted on the biochemical properties in vivo (Preston and Phizicky, 2010), we confirmed that the additional of 5′Gs has no severe effect on the experimentally determined Tm values (Appendix Fig. S1D,E). For those low yield tRNA transcripts, we introduced at least 2Gs at the 5′ end to facilitate the production efficiency. Briefly, the DNA template of each tRNA was generated using the Primerize method (Tian et al, 2015) and the T7 promoter sequence was included for in vitro transcription driven by T7 RNA polymerase (Chramiec-Głąbik et al, 2023). An overnight T7-driven transcription reaction was performed at 37 °C with the following: 40 mM Tris, pH 8.0, 5 mM DTT, 30 mM MgCl$_2$, 1 mM spermidine, 20 mM NTPs, linear DNA, RNasin (Thermo Fisher Scientific), T7 RNA polymerase (Thermo Fisher Scientific), and pyrophosphatase (Thermo Fisher Scientific). The Cy5-labeled CTP was added to the reaction to generate fluorescent RNA substrates (20% of CTP are Cy5-labeled which randomizes the labeled position and minimizes the labeling-related binding disruption). RNase-free DNase I (Thermo Fisher Scientific) was added to digest the DNA template, and the reaction was stopped by the addition of EDTA and protease K. The reaction was subjected to a FPLC system using a DEAE weak anion exchange column (GE) and followed by temperature-gradient-based annealing. Refolding of the RNA was carried out by heating the RNA solution to 80 °C for 2 min and slowly cooling to room temperature. The annealed RNAs were further purified using a Superdex 75 Increase gel filtration column (GE) in the buffer (20 mM HEPES, pH 7.5, 1 mM MgCl$_2$, 150 mM NaCl) and the fractions of interest were pooled, concentrated, and stored at −20 °C.

## Recombinant protein production

The ORFs of PUS enzymes were synthetically generated and cloned in suitable expression vectors for bacteria or insect cells, such as pETM30 (with N-terminal GST-His-tag) or pFastBac. However, human PUS10 ORF was cloned to pETM11 with N-terminus His-tag. To generate catalytically dead variants of PUS enzymes, a site-directed mutagenesis method was applied. PUS1, TRUB1/PUS4, PUS7 and PUS10 are known to be produced using the bacterial expression system, while PUS3 can be expressed using the insect cell expression method according to our previous work (Lin et al, 2022). Depending on the expression system, the cells were grown according to standard protocols. All plasmids, including the PUS1, TRUB1/PUS4, PUS7 and PUS10 constructs, were transformed into the strain BL21 (DE3) CodonPlus-RIL competent cells. Transformed cells were grown in TB broth and protein expressions were induced by isopropyl β-D-1-thiogalactopyranoside IPTG (1 mM) for overnight at 18 °C. Cells were collected and lysed in lysis buffer (50 mM Tris, pH 8.0, 300 mM NaCl, 5 mM DTT, 5% glycerol, 1 mM EDTA, 5 mM MgCl$_2$) containing protease inhibitors and DNase. The solution was sonicated, and the soluble part was separated from the cell debris using centrifugation. The protein containing supernatant was subject to affinity purification, such as NiNTA beads or GSTPrep purification. The eluate was collected and incubated with the glutathione S-transferase (GST) fused or Histidine (His) tagged Tobacco Etch Virus (TEV) protease overnight at 4 °C and followed by removal of TEV and GST-tag or His-tag. The solution was applied to a gel filtration step. An optional heparin step was introduced prior to the gel filtration depending on whether there is nucleic acid contamination.

## Microscale thermophoresis

Proteins at various concentrations (0.01–10 μM) were mixed with Cy5-labeled tRNA$^{Gln}$ substrate (30 nM) in a 10 μL reaction volume. The buffer contains 20 mM HEPES, 100 mM NaCl, 2 mM MgCl$_2$ and 2 mM DTT. The reactions were performed at 4 °C for 30 min to equilibrate the complex formation. The samples were subjected to capillaries for the Monolith device (NanoTemper Technologies GmbH) to determine the Kds. To determine Kds, samples were loaded into the premium capillaries (NanoTemper Technologies GmbH; cat. #MO-K025) and the run was controlled with MO.Control v1.4.4 (NanoTemper Technologies GmbH; cat. no MO-S001C). The Kds were calculated based on the measurement results ($n = 3$) using MO.Affinity Analysis v1.4.4 (NanoTemper Technologies GmbH; cat. no MO-S001A) and the binding response graphs were plotted using GraphPad Prism (GraphPad Software).

## Pseudouridylation assay and detection (primer extension)

Human PUS enzymes (5 μg for PUS1, PUS3, PUS4 and PUS7 or 25 μg for PUS10) were mixed with tRNA of interest (8 μg) in a 25-μL reaction volume in buffer containing 100 mM ammonium acetate, 100 mM NaCl, 20 mM Tris-Cl, pH 8.0, 5 mM MgCl$_2$, 5 mM DTT, incubated at 37 °C for 1 h and followed by phenol/chloroform extraction and precipitation overnight in EtOH at −80 °C (Lin et al, 2022). In the cases with PUS10 treatment, PUS10 was added last to the mixture and the incubation was performed at 37 °C for further

8 h. The tRNA was further treated with or without CMC to form specific CMC-Ψ conjugation, cleaned up again using phenol/chloroform extraction and precipitation in EtOH at −80 °C. The recovered RNAs were redissolved in 15 μL H$_2$O and 1 μL of tRNA solution was subjected to reverse transcriptions using SuperScript III (Invitrogen) with Cy5-labeled RNA-specific primers for 1 h at 50 °C and followed by Protease K treatment (37 °C for 30 min). The reverse transcribed cDNA products were then resolved in a 15% urea denaturing gel and run at 200 V for 60 min. The bands corresponding to with and without CMC-Ψ conjugations were visualized using the Bio-Rad GelDoc imaging system.

For time- course pseudouridinylation reaction, human PUS enzymes (2.5 μg) were mixed with tRNA of interest (4 μg) in a 25-μL reaction volume in the reaction buffer (100 mM ammonium acetate, 100 mM NaCl, 20 mM Tris-Cl, pH 8.0, 5 mM MgCl$_2$, 5 mM DTT), incubated at 37 °C for dedicated time, namely: 1 min, 30 min, 1 h and 4 h. Reactions were stopped by adding phenol/chloroform and following with extraction as described above.

## tRNAs isolation from human cells

HEK293T cells (Dharmacon) were cultured at 37 °C in 5% CO$_2$ in Dulbecco's modified Eagle's medium (Lonza) supplemented with 10% (v/v) FBS (EurX). Cells were cultured and collected from nine T75 flasks with 90% confluence. Cells were washed with PBS and lysed in 335 μL lysis buffer (10 mM Tris-HCl pH 7.5, 100 mM NaCl, 10 mM MgCl$_2$, 1% Triton X100, 0.5 mM DTT and 0.5% sodium deoxycholate). An equal volume of water was added to the lysate and the total RNA was isolated by three subsequent extractions of one volume of acid phenol-chloroform (Acid-Phenol:Chloroform, pH 4.5 (with IAA, 125:24:1)) and followed by final extraction with one volume of chloroform. During each extraction, the mixture was vortexed thoroughly and followed by centrifugation at $4500 \times g$ for 10 min at 4 °C. The upper aqueous phase was transferred to a new tube and the extraction was performed again. The RNA, in the upper phase, was precipitated with 0.1 volume of 3 M sodium acetate (pH 6.3), 3 volumes of 96% EtOH, and 10 μg glycogen. The solution was incubated overnight at −80 °C and RNA was spun down at $7100 \times g$ for 30 min at 4 °C. The RNA pellet was washed in 70% EtOH and air-dried for 2 min. The pellet was then dissolved in RNase-free water and subjected to total tRNA isolation. A NucleoBond AX100 column was equilibrated with 10 mL of equilibration buffer with Triton X100 (10 mM BisTris HCl, pH 6.3, 200 mM KCl, 15% EtOH, and 0.15% Triton X100). Total RNA was dissolved in 2 mL of equilibration buffer without Triton X-100 and applied to the column. The column was washed twice with 12 mL of wash buffer (10 mM Bis-Tris HCl, pH 6.3, 300 mM KCl, 15% EtOH). Bound tRNA was eluted with 12 mL of elution buffer (10 mM BisTris HCl, pH 6.3, 750 mM KCl, 1% EtOH) and followed by precipitation using 2.5 volumes of 96% EtOH and 10 μg of glycogen. The solution was incubated overnight at −80 °C and tRNA was spun down at $7100 \times g$ for 30 min at 4 °C. The pellet was washed twice in 70% EtOH and air-dried for 2 min. The tRNA pellet was dissolved in 20 μL of RNase-free water. Subsequently, bulk tRNA was subjected to isolation of specific single tRNA using Dynabeads MyOne Streptavidin C1 magnetic beads coupled with specific oligonucleotide at 7.5 μM concentration. Bulk tRNA (350 μg) was mixed with 200 μL magnetic beads with immobilized oligonucleotide in binding buffer (1.2 M NaCl,

30 mM HEPES pH 7.5, 15 mM EDTA). The sample was incubated 20 min at 65 °C with shaking (350 rpm), followed by incubation for 1 h at room temperature with shaking. tRNA was washed 3 times with washing buffer at 37 °C (100 mM NaCl, 2.5 mM HEPES pH 7.5, 1.25 mM EDTA). Elution was done first with high salt buffer (150 mM NaCl, 0.5 mM HEPES pH 7.5, 0.25 mM EDTA) 3 times for 5 min at 65 °C with shaking and then with low salt buffer (100 mM NaCl, 0.5 mM HEPES pH 7.5, 0.25 mM EDTA) 3 times for 5 min at 65 °C with shaking. Elution fractions were combined and precipitated with 2.5 volumes of 96% EtOH at −80 °C. The sample was spun down at $7100 \times g$ for 30 min at 4 °C and washed twice with 70% EtOH. The pellet was dissolved in 15 μL RNase-free water. The quality of extracted tRNA was checked by resolving the samples in a 10% urea gel, staining with ethidium bromide solution, and visualizing with a BioRad ChemiDoc imaging system.

Thiolated tRNAs were separated from non-thiolated tRNAs on a 10% denaturing poly-acrylamide gel (7 M urea, 0.5% TBE) supplemented with 130 μM [N-acryloylamino]phenyl)mercuric chloride (APM) (Sokołowski et al, [2024]). APM interacts with exposed thiol groups and leads to an increased retardation of thiolated tRNAs in comparison to non-thiolated tRNAs. In gels without APM thiolated and non-thiolated tRNA are not separated and run at the same migration speed.

## Determination of tRNAs thermal stability changes

The thermal stability of the tRNAs was evaluated with the use of a Quant-iT RiboGreen™ RNA fluorescent probe (RiboGreen) (R11491 ThermoFisher Scientific). 200 nM of tRNA produced in-house was mixed with 200x diluted RiboGreen probe in 50 μL of the assay buffer containing 20 mM HEPES, pH 7.5, 150 mM NaCl, 1 mM MgCl$_2$, 5 mM DTT. The assay volume was mixed, short-spined, and incubated for 10 min at room temperature protected from light. 10 μL of the assay mix was loaded into capillaries (NanoTemper Technologies GmbH, cat. no AN-041001)—a total of 4 capillaries per the condition set. The capillaries were sealed with oil-based liquid rubber (NanoTemper TechnologiesGmbH, cat. no PR-P001). Capillaries were loaded into the Andromeda system from NanoTemper Technologies, allowing monitoring the fluorescence signal in the temperature gradient. The data was collected with AN.Control Software v1.1 (NanoTemper TechnologiesGmbH, cat. no AN-020001). The fluorescence excitation was adjusted to maintain total intensity at 510 nm below 2000 counts. The temperature ramp was set to 1 °C/min and fluorescence signal was collected for the temperature range 25–95 °C. Collected raw data were exported, processed and plotted (box plot) with Origin Pro 2023 (OriginLabs). For the line plots representing change of the fluorescence signal as a function of temperature the signal was further processed for comparisons across conditions (different tRNAs, PUS enzyme reactions). The signal processing involved linear interpolation to obtain equal size of the temperature vector, Min-Max normalization of the fluorescence signal, and first derivative determination with baseline correction using PeakAnalyzer from OriginLabs.

## tRNA site-specific thermal stability analysis

tRNA$^{Gln}_{UUG}$ labeled with a Cy5 was obtained from Metabion (Metabion International AG, Germany). The Cy5 label was placed at the 5′ end and U$_{48}$ position of the tRNA. The lyophilized tRNAs

were resuspended in miliQ grade water at 100 µM. Before the enzymatic reaction, tRNA refolding was carried out by heating the RNA solution at 80 °C for 2 min and slowly cooling to room temperature. A 10 nM solution of enzymatically modified tRNAs was prepared in 100 µL of 20 mM HEPES, pH 7.5, 150 mM NaCl, 1 mM $MgCl_2$, 5 mM DTT. The assay volume was mixed, short-spined, and incubated for 10 min at room temperature. 10 µL of the solution was loaded into capillaries (NanoTemper Technologies GmbH, cat no. AN-041001)—a total of 4 capillaries per condition set. Thermal unfolding analysis was performed with the use of Andromeda system (NanoTemper Technologies GmbH) using Cy5 as a source of fluorescence signal; run parameters and data analysis were performed as described above for the Quant-iT Ribogreen™ probe.

## Analysis of PUS enzymatic reaction efficiency through tRNA thermal stability

The tRNA of interest, modified individually with PUS4, PUS7, and the combination PUS3,4,7, was pre-diluted to 1 µM in 30 µL of the assay buffer comprising 20 mM HEPES (pH 7.5), 150 mM NaCl, 1 mM $MgCl_2$, and 5 mM DTT. Subsequently, titrations of PUS4 against PUS-3/4/7 and PUS7 against PUS-3/4/7 were prepared in separate tubes according to the provided tables. The total molar concentration of tRNA was maintained at 250 nM, with a total volume of 40 µL in the assay buffer. In each tube, 10 µL of a 40x pre-diluted RiboGreen probe was added, resulting in a final concentration of the tRNA of interest at 200 nM, with the fluorescent probe being 200x diluted. As a reference, 200 nM of non-modified tRNA of interest was mixed with a 200x diluted RiboGreen probe in 50 µL. This reference was employed to calculate ΔTms during the subsequent analysis. The reactions were incubated for 10 min at room temperature with RiboGreen, followed by the determination of tRNA stability using the PR.Andromeda system, as previously described.

## Analysis of the tRNA thermal stability pattern endurance

Non-modified tRNA, tRNA incubated with PUS-3/4/7, and tRNA treated with inactive mutants of PUS-3/4/7 enzymes were separately mixed with a RiboGreen probe using the following procedure. A 200 nM concentration of in-house-produced tRNA was combined with a 200x diluted RiboGreen probe in a 50 µL assay buffer containing 20 mM HEPES (pH 7.5), 150 mM NaCl, 1 mM $MgCl_2$, and 5 mM DTT. The assay mixture was thoroughly mixed, briefly spun, and then incubated for 10 min at room temperature, protected from light. Subsequently, 10 µL of the assay mix was loaded into capillaries (NanoTemper Technologies GmbH, cat. no AN-041001), with a total of 4 capillaries per condition. These capillaries were sealed with oil-based liquid rubber (NanoTemper Technologies GmbH, cat. no PR-P001) and then inserted into the Andromeda system from NanoTemper Technologies. This system allowed for the monitoring of the fluorescence signal in a temperature gradient. Data collection was performed using AN.Control Software v1.1 (NanoTemper Technologies GmbH, cat. no AN-020001). The fluorescence excitation was adjusted to maintain a total intensity at 510 nm below 2000 counts. The temperature ramp was set to 1 °C/min, and the fluorescence signal was collected over the temperature range of 25–95 °C in three

consecutive cycles of unfolding-refolding, with continuous monitoring of fluorescence changes. The analysis of the melting profiles and ΔTm calculations were conducted as described in the section on the determination of tRNA thermal stability changes.

## Isothermal and thermal unfolding size analysis of tRNAs

The hydrodynamic radius of tRNAs was assessed with a Prometheus Panta nanoDLS module (NanoTemper Technologies GmbH). 20 µM of in-house produced tRNA was prepared in 30 µL of 20 mM HEPES, pH 7.5, 150 mM NaCl, 1 mM $MgCl_2$, 5 mM DTT. The assay volume was mixed, short-spined, and incubated for 10 min at room temperature. 10 µL of the solution was loaded into the capillaries (NanoTemper Technologies GmbH, cat no. PR-C006)—a total of 3 capillaries per condition set. The capillaries were loaded onto the Prometheus Panta system and sealed with oil-based liquid. Data were collected using PR.Panta Control software v1.4.4 (NanoTemper Technologies PR-S061). For the isothermal size analysis of tRNA the nanoDLS run was set to 5 s and 10 acquisitions per capillary. The hydrodynamic radius of tRNA was determined using the regularization model implemented in the software and the intensity plots were exported. For the thermal unfolding analysis of the tRNA hydrodynamic radius the nanoDLS run was set to a 500 ms acquisition per capillary. The thermal ramp was set at 1 °C/min and the temperature analysis range was adjusted to 20–95 °C. Hydrodynamic radius changes in function of temperature were determined using the cumulant radius model implemented in the software. The raw data were exported, analyzed and plotted using OriginLabs. Predictions of rH based on the tRNA molecular weight were calculated using available online converter (www.fluidic.com/toolkit/hydrodynamic-radius-converter/).

## Electron microscopy

QUANTIFOIL® R 2/1 copper grids (200 mesh) were glow discharged on a Leica EM ACE 200 glow discharger (8 mA, 60 s). 2.5 µL of 15 µM tRNA in assay buffer (20 mM HEPES, pH 7.5, 150 mM NaCl, 1 mM $MgCl_2$, 5 mM DTT) was plunge-frozen using a Vitrobot Mark IV (Thermo Fisher Scientific) set to 100% humidity and 4 °C with the following blotting parameters: wait time 1 s, blot force 5 s and blot time 2 s. The datasets were collected on a 300 keV Titan Krios G3i (Thermo Fisher Scientific, Solaris, Poland) equipped with a Gatan BioQuantum energy filter and a Gatan K3 BioQuantum direct electron detector. The under-focus range was −0.8 to −2.4 µm for a total of 40 frames accumulating an overall dose of 40 e−/Å². Respective numbers of collected micrographs for each dataset can be found in the Table 1. The used pixel size was 0.415 Å for in vivo tRNA$^{Arg}_{UCU}$ and 0.86 Å for the remaining datasets.

## Image processing

All micrographs were subjected to motion correction using WARP5 with subsequent CTF estimation. Then the micrographs were subjected to the selection protocol to fish out only the best quality images for further analysis. This task was done using the ExposureCuration job in cryoSPARC with a set of parameters as follows: image resolution <8 A, defocus tilt angle <20°, astigmatism <500 nm. Selected micrographs were subjected to further

processing steps. Picked particles together with averaged micrographs were imported to cryoSPARC and analyzed in detail. As a first step, reference free 2D classification was used to select the best possible particles and create templates for the template picker. As a validation of selected template, template picker was always run on a small subset of micrographs (typically 500-1000). Template-picked particles were 2D classified in reference-free 2D classification(s) until best particles were selected (typically 3-to-5 rounds). The particles from the best 2D classes (resolution $\leq 8$ Å, ECA $\leq 3$) were used to build initial 3D models, from which the one contains all essential features was used to train TOPAZ (Bepler et al, 2019) to increase the number of good particles that could have been missed out during previous steps. TOPAZ picked particles were submitted to few rounds of reference free 2D classification to clean the particles set. The particles with all required density were used to create new 3D models, of which only the best one containing all essential features was refined and determined the solution. To refine the best model, the Homogenous Refinement (HomoRef) (Punjani et al, 2017) followed by Non-Uniform refinement (Punjani et al, 2020) protocols were used. The FSC correlation curve was calculated for each final reconstruction using a gold standard FSC = 0.143 threshold. Local resolution was calculated in LocalResolutionEstimation job implemented in cryoSPARC and implements a local windowed FSC method for estimating local resolution, similar to the blocres program (Cardone et al, 2013).

## tRNA 3D structure determination and modeling

The initial models of human tRNA 3D structures were built using comparative modeling with ModeRNA (Rother et al, 2011), using a collection of experimentally determined tRNA structures as templates (with RCSB PDB code 6r7q_A as the main template). These models were superimposed onto the corresponding cryo-EM maps, with the aid of the Fit in Map tool in ChimeraX (Pettersen et al, 2021). The models were flexibly refined in the context of the cryo-EM maps using a coarse-grained modeling tool SimRNA (Boniecki et al, 2015), which performs Monte Carlo simulations and evaluates RNA conformations with a statistical potential. For the cryo-EM tRNA structure determination, distance restraints were used to maximize the preservation of canonical and non-canonical base-pairs and stacking interactions conserved in tRNAs. 10 models that scored best according to the SimRNA scoring function had their all-atom representation reconstructed. These models were further refined against the map using Phenix real_space_refine (Afonine et al, 2018) with macro_cycles=1 and restraints derived from the initial model using doubleHelix (Chojnowski, 2023). Finally, we used QRNAS (Stasiewicz et al, 2019) to minimize steric clashes and improve local geometry. The fit of models to the map was assessed using QScore with the MapQ plugin (Pintilie et al, 2020) in ChimeraX.

## Computational analysis of tRNA thermal stability

Starting with the folded 3D structure of tRNA$^{Gln}_{UUG}$, a series of isothermal single-replica SimRNA simulations were run, without any restraints, to assess the stability of the global tRNA structure and the pairwise contacts made by individual ribonucleotide residues. The probability of accepting or rejecting a move in the SimRNA simulation is based on a Metropolis criterion (Metropolis et al, 1953), which depends on the T parameter, analogous to the physical concept of temperature in statistical mechanics. If the new state in the SimRNA Monte Carlo simulation corresponds to a higher (less favorable) energy, it is accepted with a probability governed by the Boltzmann factor, exp(-ΔE/kT), where ΔE is the difference in energy between the two states, k is the Boltzmann constant, and T is the temperature. Therefore, the temperature parameter T modulates the probability of accepting higher-energy states, and consequently, it controls the degree of exploration of the conformational space. A higher temperature allows for more frequent acceptance of higher-energy states, thereby enabling a broader exploration of the conformational space, while a lower temperature narrows this exploration to energy minima, by prioritizing lower-energy states. To simulate the structural stability of tRNA$^{Gln}_{UUG}$, we used T ranging from 0.6 (where the tRNA remained folded and very similar to the structure determined by cryo-EM) to 1.4 (where the tRNA unfolded quickly). For all frames from each simulation, all-atom models were reconstructed and compared with the cryo-EM structure as a reference. For every residue in each conformation, pairwise contacts were identified and compared with the reference structure using 1D2DSimScore (Moafinejad et al, 2023), to determine whether a given residue retains native-like interactions or if its native-like contacts are broken. The hydrodynamic radii of the models were assessed using HullRad (Ortega et al, 2011; Fleming and Fleming, 2018). To compare hydrodynamic radii from simulations with the experimental data, the simulated radii values were recalibrated by subtracting the factor of 0.5 nm that accounts for the underestimation of the hydrodynamic radii by the cumulant radius model. All values were exported to xls files.

## Molecular dynamics simulations

Molecular dynamics (MD) simulations were performed for four RNA variants of tRNA$^{Gln}_{UUG}$: without any modifications, with single modifications $\Psi_{13}$ or $\Psi_{55}$, and both $\Psi_{13}$ and $\Psi_{55}$. The structure of tRNA$^{Gln}_{UUG}$ built into the cryo-EM density map was used as the starting model. The modifications were introduced using ModeRNA (Rother et al, 2011). The system was prepared using the Amber χOL3 force field (Pérez et al, 2007; Zgarbová et al, 2011; Banáš et al, 2010), and the modified nucleotide parameters for Ψ were generated using Antechamber (Wang et al, 2006, 2004). Both systems were solvated in a TIP3P water (Sun and Kollman, 1995) box with at least a 10 Å buffer, and counterions were added for neutralization. Initial minimization was performed in two stages to relieve any steric clashes and relax the system's energy. Heating was conducted using a Langevin thermostat to gradually raise the system temperature from 0 K to 300 K under constant volume conditions, with heavy atom restraints applied to maintain RNA structural integrity.

Equilibration followed in three stages, each with progressively reduced positional restraints, allowing the system to adjust to the target temperature and pressure. Constant pressure periodic boundary conditions were used to mimic physiological conditions, with a pressure relaxation time of 2 ps. After withdrawing all constraints, 1 ns MD simulations were carried out under the conditions of constant temperature and constant volume (NVT) as well as constant temperature, constant pressure (NPT). In NPT condition, a pressure of 1.0 atm was kept using the Berendsen barostat (Berendsen et al, 1984) with a relaxation time of 2 ps. The

SHAKE (Ryckaert et al, 1977) algorithm was employed to constrain all bond lengths involving hydrogen atoms. The final production run, conducted for 500 ns, utilized the GPU-accelerated pmemd.-cuda engine of AMBER22 (Götz et al, 2012) with a 2 fs time step. The choice of parameters, such as Langevin thermostat (Turq et al, 1977) ($\gamma = 1 \text{ ps}^{-1}$) and particle mesh Ewald (PME) (Darden et al, 1998) or long-range electrostatics, was made to balance computational efficiency and accuracy, ensuring stable temperature and pressure control throughout the simulation. Three independent runs for each RNA variant were calculated to mitigate the potential influence of random initial conditions and stochastic fluctuations that could arise during the simulations. MD trajectories were analyzed by cpptraj (Roe and Cheatham, 2013). Stacking energies were calculated using the LIE (de Amorim et al, 2008) utility of AmberTools (Case et al, 2023).

### Quantifications and statistical analyses

All details related to statistical analysis can be found in figure legends. Graphed datasets are represented in box plots. The central line inside the box indicates the median and the square the mean. The lower and upper edges correspond to the first (Q1) and third (Q3) quartiles, respectively, defining the interquartile range (IQR). The whiskers extend to the minimal and maximal values within the 1.5IQR. Points beyond this range are considered outliers and are shown as transparent points. All statistical analyses were performed using one-way ANOVA ($\alpha = 0.05$) with a Bonferroni multiple comparisons test. The exact $p$ values are shown in each panel.

## Data availability

The micrographs and cryo-EM densities have been deposited in the Electron Microscopy Public Image Archive (EMPIAR) and the Electron Microscopy Data Bank (EMDB) with the following accession codes—$\text{tRNA}^{\text{Gln}}_{\text{UUG}}$ (unmodified EMPIAR-11514; EMD-16940; $\Psi_{13}/\Psi_{39}$–modified EMD-16939; $\Psi_{13}/\Psi_{55}$–modified EMPIAR-11515, EMD-16941), $\text{tRNA}^{\text{Gly}}_{\text{CCC}}$ (unmodified-EMPIAR-11516, EMD-16943; $\Psi_{13}/\Psi_{55}$–modified EMPIAR-11517, EMD-16945; $\Psi_{13}/\Psi_{39}/\Psi_{55}$–modified EMD-16946), $\text{tRNA}^{\text{Glu}}_{\text{UUC}}$ (unmodified EMD-16947, $\Psi_{13}/\Psi_{55}$–modified EMPIAR-11518, EMD-16948), $\text{tRNA}^{\text{Asp}}_{\text{GUC}}$ (unmodified EMD-16950, $\Psi_{13}/\Psi_{55}$–modified EMPIAR-11519, EMD-16951). The dynamically fitted models (https://doi.org/10.6084/m9.figshare.24871854.v1), SimRNA simulations data (https://doi.org/10.6084/m9.figshare.23560173) and the trajectories of the MD simulations (https://zenodo.org/records/13814811) have been deposited at publicly available repositories. The raw data for all biophysical measurements are available (see Data S1). This paper does not report original code.

The source data of this paper are collected in the following database record: biostudies:S-SCDT-10_1038-S44318-025-00443-y.

## Peer review information

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

## Acknowledgements

Research funding for this study was provided by European Research Council (ERC) under the European Union's Horizon 2020 research and innovation program grant No. 101001394 (SG), Polish National Science Center grant 2020/37/B/NZ2/02456 (JMB), Polish high-performance computing infrastructure PLGrid (HPC Centers: ACK Cyfronet AGH) grants PLG/2020/014021, PLG/2021/014959 and PLG/2022/015771 (ADB), and PLG/2023/016901 (Su. M). We would like to thank Grzegorz Wazny and the whole Solaris team for their constant support during cryo-EM grid screening and sample optimization. In addition, we are grateful to all members of the Glatt lab for insightful discussion during the preparation of this manuscript. The work is supported under the Polish Ministry and Higher Education project: "Support for research and development with the use of research infrastructure of the National Synchrotron Radiation Centre SOLARIS" under contract nr 1/SOL/2021/2.

## Author contributions

**Anna D Biela**: Data curation; Formal analysis; Validation; Investigation; Visualization; Methodology; Writing—original draft; Writing—review and editing. **Jakub S Nowak**: Data curation; Formal analysis; Validation; Investigation; Visualization; Methodology; Writing—original draft; Writing—review and editing. **Artur P Biela**: Data curation; Software; Formal analysis; Investigation; Visualization; Writing—original draft; Writing—review and editing. **Sunandan Mukherjee**: Software; Formal analysis; Validation; Investigation; Visualization. **Seyed Naeim Moafinejad**: Software; Formal analysis; Validation; Investigation; Visualization. **Satyabrata Maiti**: Software; Formal analysis; Validation; Investigation; Visualization. **Andrzej Chramiec-Głąbik**: Formal analysis; Investigation; Methodology. **Rahul Mehta**: Formal analysis; Validation; Investigation. **Jakub Jeżowski**: Formal analysis; Validation; Investigation. **Dominika Dobosz**: Formal analysis; Validation; Investigation; Methodology. **Priyanka Dahate**: Formal analysis; Validation; Investigation; Methodology. **Veronique Arluison**: Formal analysis; Validation; Investigation. **Frank Wien**: Formal analysis; Validation; Investigation. **Paulina Indyka**: Data curation; Investigation. **Michal Rawski**: Data curation; Investigation. **Janusz M Bujnicki**: Conceptualization; Resources; Data curation; Software; Formal analysis; Supervision; Funding acquisition; Validation; Investigation; Visualization; Writing—original draft; Writing—review and editing. **Ting-Yu Lin**: Conceptualization; Data curation; Formal analysis; Supervision; Validation; Investigation; Visualization; Methodology; Writing—original draft; Writing—review and editing. **Sebastian Glatt**: Conceptualization; Resources; Data curation; Formal analysis; Supervision; Funding acquisition; Validation; Investigation; Visualization; Methodology; Writing—original draft; Project administration; Writing—review and editing.

Source data underlying figure panels in this paper may have individual authorship assigned. Where available, figure panel/source data authorship is listed in the following database record: biostudies:S-SCDT-10_1038-S44318-025-00443-y.

## Disclosure and competing interests statement

The authors declare no competing interests.

# Expanded View Figures

**Figure EV1.   Molecular simulations of tRNA unfolding.**

(**A**) NanoDLS analysis of tRNA$^{Gln}_{UUG}$ size changes in the thermal gradient. 2.28 nm rH corresponds to the simulated start of the unfolding process, 2.65 nm rH corresponds to the experimentally calculated inflection point, 3.82 nm rH corresponds to the average value of simulated fully unfolded tRNA$^{Gln}_{UUG}$. (**B**) Conformational space sampled by SimRNA at the different values of the T (temperature) parameter, from very "cold" (0.6) sampling of near-native structures to very "hot" (1.4) sampling of unfolded conformations (left). Relationship between the deviation of the progressively unfolded RNA 3D conformation from the folded structure and the rH value (right). (**C**) Selected models representing each of tRNA unfolding states. Light blue indicates the D-arm, yellow indicate T-arm. (**D**) Calculated hydrodynamic radii of simulated different unfolding states of the tRNA$^{Gln}_{UUG}$ (top). Summary table of mean hydrodynamic radii calculated from the simulated unfolding states of tRNA$^{Gln}_{UUG}$ and their recalibrated values (bottom). (**E**) A heatmap representing the frequency of base-pairing retention in SimRNA simulations at the T parameter ranging from 0.6 to 1.4. Source data are available online for this figure.

►

                                              

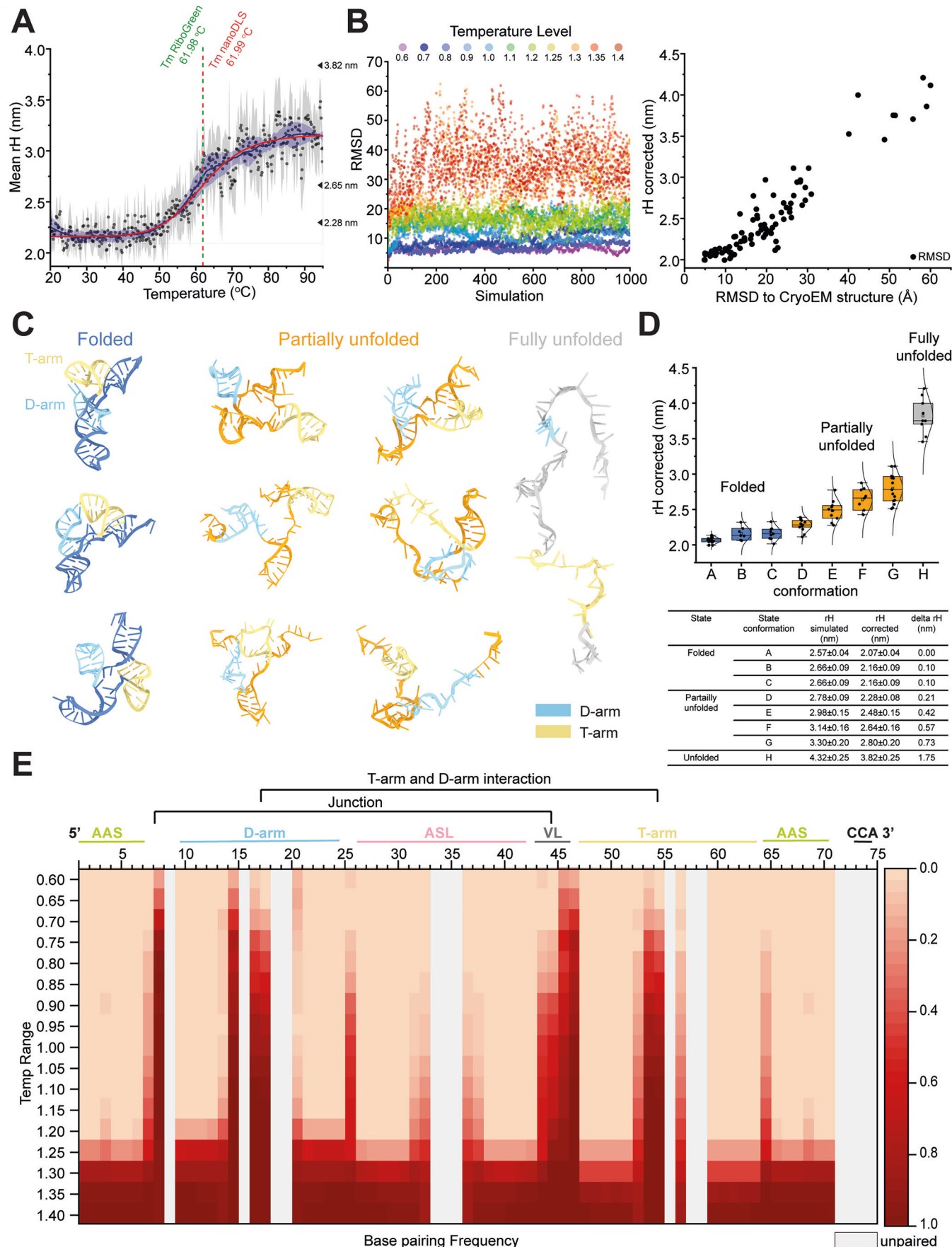

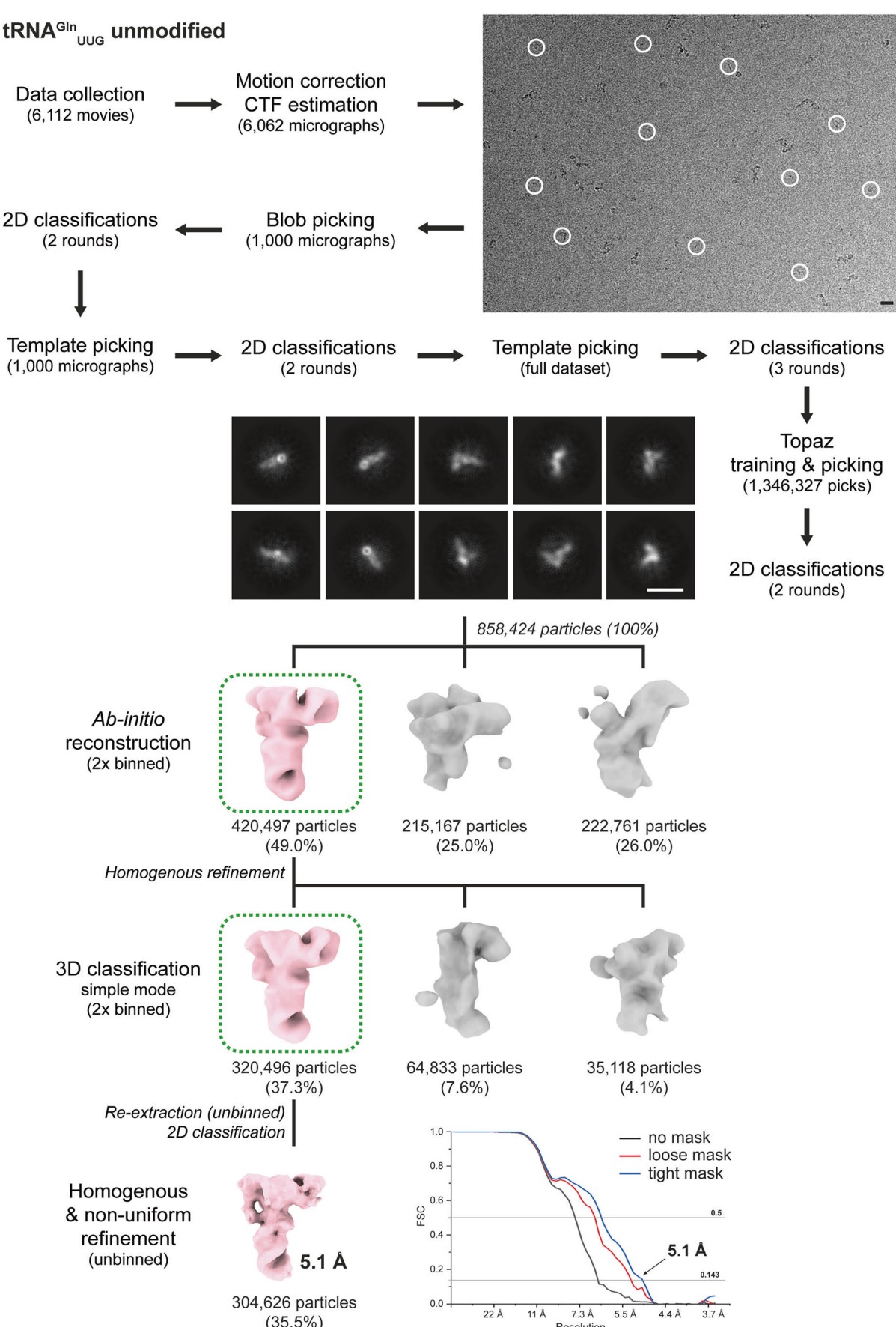

**Figure EV2. Cryo-EM reconstruction pipeline for unmodified tRNA$^{Gln}_{UUG}$.**

A representative micrograph is shown (top right) together with indicated positions of the finally selected particles (white circles); scalebar = 100 Å. Representative 2D classes, ab initio classes and further steps of 3D refinement are shown in the bottom part of the figure. Absolute numbers and percentages of particles are listed and the Fourier Shell Correlation blot (FSC) blot of the final reconstruction, highlighting the nominal resolution at FSC = 0.143.

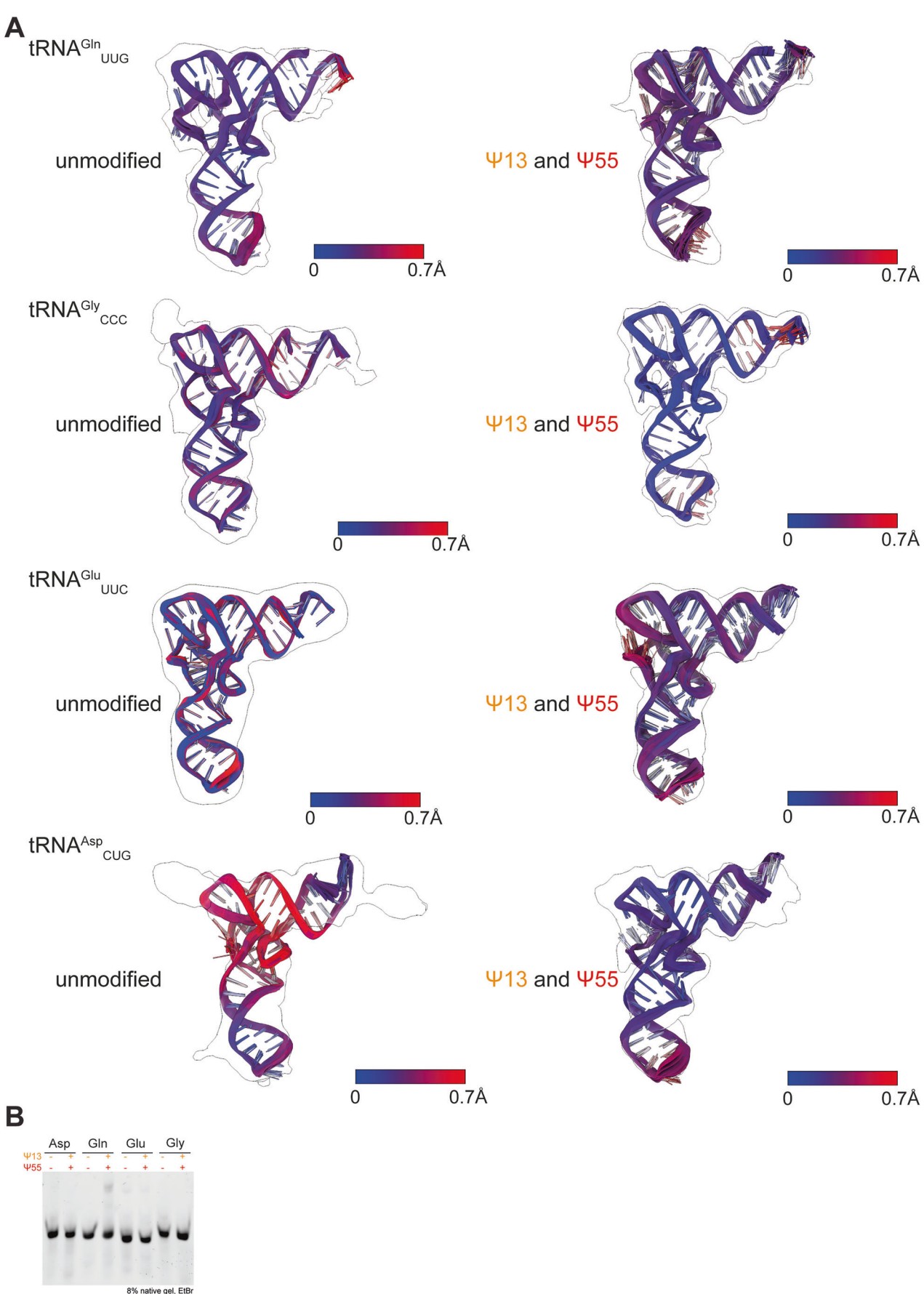

**Figure EV3. Comparison of the obtained models within each ensemble.**

(A) From top to bottom. Cryo-EM reconstructions of unmodified (left) and $\Psi_{13}$- and $\Psi_{55}$-modified (right) human RNA$^{Gln}_{UUG}$, tRNA$^{Gly}_{CCC}$, tRNA$^{Glu}_{UUC}$ and tRNA$^{Asp}_{GUC}$ with an ensemble of 10 atomic models fitted into the density colored by RMSD between each atom. (B) A native gel showing the mobility of unmodified and $\Psi_{13}$- and $\Psi_{55}$-modified tRNAs.

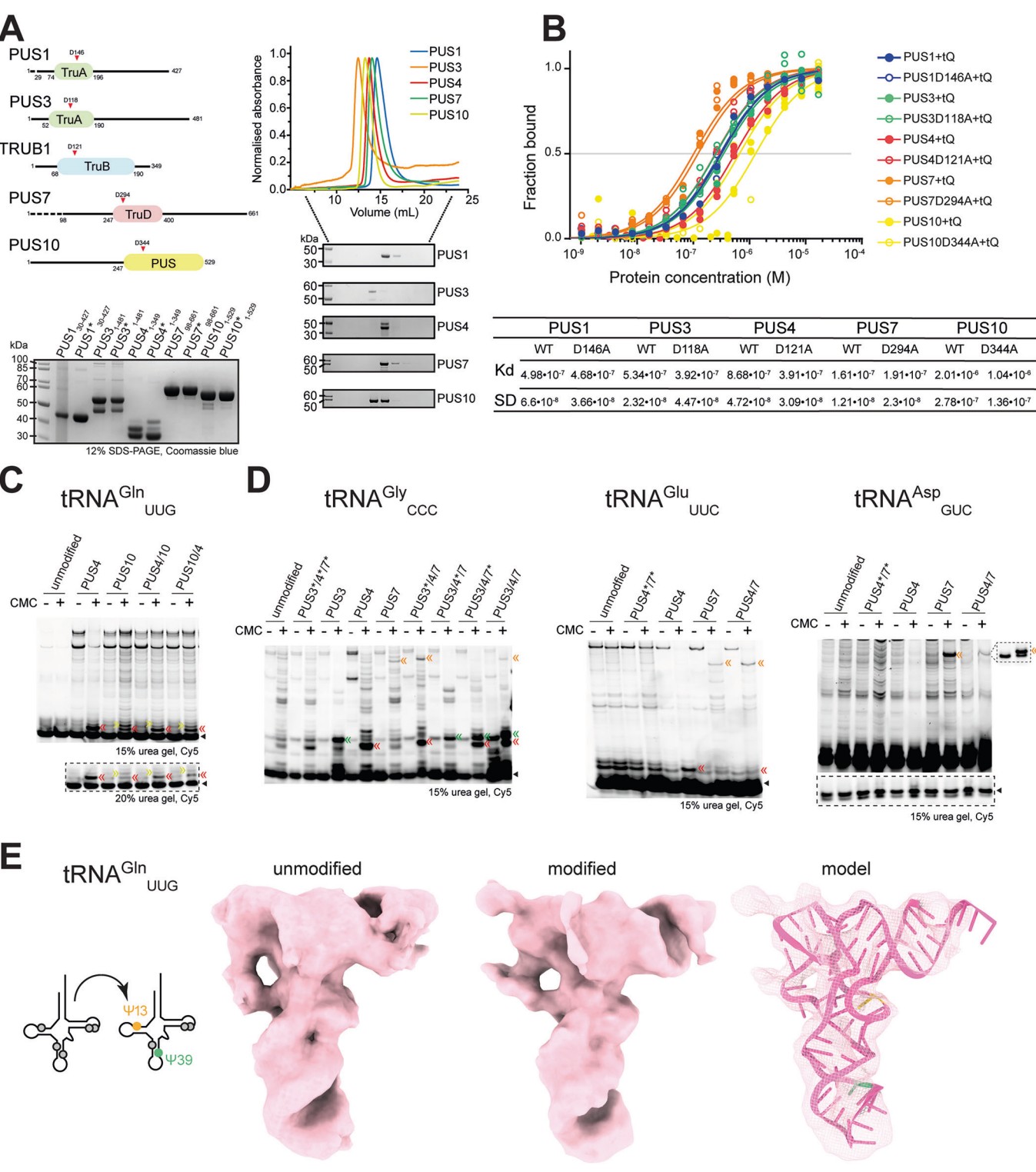

**Figure EV4.   In vitro pseudouridinylation of human tRNAs.**

(A) Cartoon presentation of domains of each PUS (upper-left). The length is labeled, and domain is highlighted where the catalytic residue is indicated by a triangle. The truncated regions of PUS1 and PUS7 are shown as dashed lines. An SDS-PAGE gel of recombinant *Hs*PUS enzymes (lower-left). Catalytic mutants marked with stars. SEC profiles of purified human PUS enzymes (upper-right) and the collected fraction for each purification were resolved in SDS-PAGE gels (lower-right). (B) MST analyses of PUS, including wild type and inactive form (DA), binding to $tRNA^{Gln}_{UUG}$. The calculated Kds of wild-type PUS and inactive forms (DA), binding to $tRNA^{Gln}_{UUG}$. (C, D) Detection of PUS-dependent Ψ formation on $tRNA^{Gln}_{UUG}$, $tRNA^{Gly}_{CCC}$, $tRNA^{Glu}_{UUC}$ and $tRNA^{Asp}_{GUC}$. The reverse-transcribed cDNA products were resolved in a 15% or 18% urea gel and the CMC-Ψ mediated short cDNAs are indicated by double-arrows ($\Psi_{13}$ orange, $\Psi_{27/28}$ blue, $\Psi_{39}$ green, $\Psi_{54}$ yellow $\Psi_{55}$ red). Each tRNA primer (labeled with Cy5) is indicated by a triangle. In the case of $tRNA^{Asp}_{GUC}$, the signal for primer is obtained from a short exposure shown in the dash lined box while the $\Psi_{13}$-dependent cDNA is obtained using a site-specific primer (shown in a dash lined box on the side). (E) Cryo-EM reconstructions of unmodified and $\Psi_{13}$- and $\Psi_{39}$-modified $tRNA^{Gln}_{UUG}$. The modified sites are highlighted by color code in the 2D cartoon and the model ($\Psi_{13}$ orange and $\Psi_{39}$ green). Source data are available online for this figure.

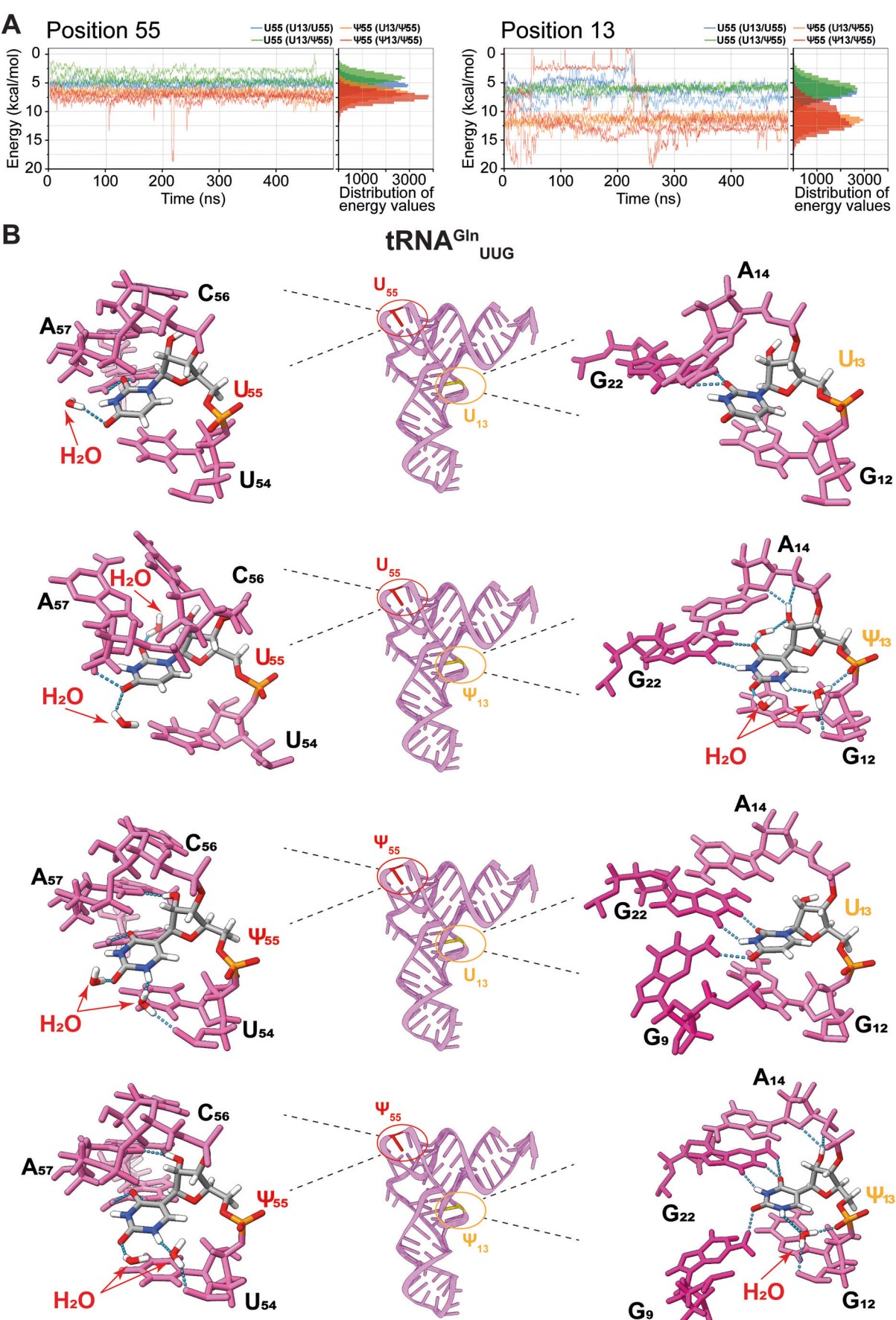

**Figure EV5. MD Simulations of tRNA with and without $\Psi_{13}$ and $\Psi_{55}$.**

(A) Distribution of interaction energies between uridine (U) and pseudouridine ($\Psi$) with neighboring residues throughout the simulation and a histogram of the energy distribution for $U_{55}/\Psi_{55}$ (left) and $U_{13}/\Psi_{13}$ (right). The graphs emphasize the varying stability and dynamics of interactions across different simulation variants. Notably, variants containing $\Psi$ (depicted in orange and red) exhibit greater stability compared to those with U, represented in green and blue. (B) Snapshots illustrating the local environments at positions 55 (left) and 13 (right) for all simulation variants. In the simulations where position 13 is occupied by $\Psi$ (specifically in $\Psi_{13}/U_{55}$ and $\Psi_{13}/\Psi_{55}$), a water molecule (red arrow) forms a triad of hydrogen bonds involving the imino hydrogen of $\Psi_{13}$ and the phosphate groups of both $\Psi_{13}$ and $G_{12}$. At position 55, when $\Psi_{55}$ is present (in $U_{13}/\Psi_{55}$ and $\Psi_{13}/\Psi_{55}$), a water molecule (red arrow) bridges between the imino hydrogen of $\Psi_{55}$ and the OP of $U_{54}$. Notably, these specific interactions are absent when U occupies both the position 13 and 55.

