## [Peer Review File · The EMBO Journal]

Determining the effects of pseudouridine incorporation on human tRNAs

Anna Biela, Jakub Nowak, Artur Biela, Sunandan Mukherjee, Seyed Moafinejad, Satyabrata Maiti, Andrzej Chramiec-Głabik, Rahul Mehta, Jakub Jeżowski, Dominika Dobosz, Priyanka Dahate, Veronique Arluison, Frank Wien, Paulina Indyka, Michal Rawski, Janusz Bujnicki, Ting-Yu Lin, and Sebastian Glatt

Corresponding author(s): Sebastian Glatt (sebastian.glatt@uj.edu.pl) , Janusz Bujnicki (janusz@iimcb.gov.pl), Ting-Yu Lin (ting-yu.lin@durham.ac.uk)

Review Timeline:

Submission Date:	10th Jan 25
Editorial Decision:	10th Mar 25
Revision Received:	18th Mar 25
Accepted:	2nd Apr 25

Editor: Cornelius Schneider

Transaction Report:

This manuscript was transferred to The EMBO JOURNAL following peer review at another journal.

Reviewer #1:

The authors provide a manuscript describing the influence of pseudouridylation on tRNA stability. They use T_m measurements, cryo-EM and MD simulation to probe tRNA structure and dynamics.

In the first part of the manuscript, they report the production of 10 in vitro transcribed (IVT) tRNAs of which only 8 are stable. They perform MD simulations on one tRNA and confirm the G19-C56 base pair as essential for tRNA folding. This base pair is invariant and responsible for the structural integrity of tRNAs; mutations of the G19-C56 have served in multiple studies to disrupt tRNA tertiary structure.

Next, the authors perform SPA cryo-EM on 2 of the 8 IVT tRNAs; the L-shape is visible, the phosphodiester backbone, the major and minor grooves can be distinguished, but the bases are not resolved. The resolution seems reasonable for such a small particle. But the number value of a given resolution is only relevant in the context of an interpretation, which I miss here. For example, in this case, could tRNA species be distinguished on the basis of their reconstructions?

Next follows a biochemical/biophysical analysis of pseudouridylation and its effect on tRNA stability and dynamics. This is the heart of the study and not reflected in the title. All bacterial pseudouridylation enzymes were purified and used to modify IVT tRNA in different combinations, leading to the identification of specific modification sites that are important for tRNA stability. I am not sure whether cryo-EM (more specifically here the interpretation of sample resolution, local resolution, or sample stability/visibility on grids) is a good readout for sample dynamics. The lack of relation between sample quality and biological interpretation is something the authors themselves state later in the manuscript, "The obtainable resolution (~ 10 Å) for the endogenous tRNAs was limited in comparison to the well-defined maps of the in vitro-derived tRNAs, and the reasons could be manifold." Especially, in one case the cryo-EM map quality and T_m values are contradictory. In-solution techniques (SAXS, NMR, T_m measurements) would be more informative for tRNA dynamics. It would be great to see a comparison of several tRNA species directly comparing IVT (unmodified)– IVT PUS enzyme treated (partially modified) – endogenous (fully modified).

Since pseudouridylation does not interfere with the canonical base-pair interface, via which mechanism does it stabilize the tRNAs? Could MD give answers? Why do certain positions stabilize more than others? I miss the rationale for why hyper-pseudouridylation causes the inverse effect.

Reviewer #2

This is a nice experimental and theoretical study, using several techniques, on various human tRNAs. Seeing unmodified and variously modified tRNAs by cryo-EM is an accomplishment, even despite the low resolution around 5.0Å (which brings us back to the early 1970's using X-ray crystallography but, here, we have structural analyses of 6 human tRNAs!). The low resolution is compensated by a systematic application of established techniques like thermal melting, molecular dynamics simulations, and especially very elegant biochemical experiments using specific modifying enzymes, each one targeted to a known position in the tRNA sequence for various tRNA sequences. The paper illustrates biochemically and structurally in detail how each tRNA sequence has unique molecular properties, a fact that is often forgotten or poorly appreciated. The paper has important consequences on the biochemical mechanisms of PUS enzymes and on tRNA folding (for example, the key role of Ψ13 for native folding).

I do not have many points to raise and to comment on.

Line 50. "decorated" a common but pejorative use for tRNA modifications; a "decoration" is not functionally necessary, which is contrary to the main point of this paper that shows how subtle the tRNA modifications can be for tRNA folding.

Line 55. What is known about the relationships between pseudouridine modifications in tRNAs and aminoacylation? Are some of the positions studied involved in tRNA recognition by aaRSs?

Line 99; "intrinsically folded"? Could the authors clarify what they intend to mean. Simple 2D predictions show that tRNA-Gln folds directly into the standard cloverleaf (free energy equals to -26 kcal/mol, using mfold) while tRNA-Pro does not (with a "better" free energy -28 kcal/mol). It would have been nice to have such calculations on the studied tRNAs (ideally with comparative plots of the partition plots) presented, at least in the supplementary material. It has been noticed for a long time that the calculated 2D structure of some tRNA sequences give as the minimal energy the cloverleaf, while others do not (therefore the native folding should depend more on targeted specific modifications). In this respect, native gels of the unmodified transcripts would also have been instructive (ideally with some additional modified positions).

These experiments are easy to perform, but assuredly only potentially indicative.

Lines 155-159. What is known about the sequence specificity of the PUS enzymes? For example, in both tRNA-Asp and tRNA-Gly, U13 becomes modified but not U25? There are many other examples clearly.

Line 163. Ψ54 is rare in eukaryotes (see <https://doi.org/10.3390/genes12020140>). The authors should cite the precise references for the presence of modifications in the tRNAs they measured. In Modomics, if I am not mistaken, I do not see a Ψ54 in tRNA-Gln. Ψ54 is mainly seen in Archaea and among them in (hyper)thermophiles.

Line 233. In some eukaryotic tRNA-His, 54 is modified into pseudouridine, but not position 55 according to Modomics. Can the authors give the original references please? Or is this modification identified in this paper?

Line 235. Merriam-Webster Online says that neglectable is an archaic form of negligible. Some other dictionaries omit neglectable entirely.

Line 315 (with Lines 155-159). Wouldn't this imply that there is a co-evolution between tRNA sequences and modification enzymes? See also Line 55.

Reviewer #3

The manuscript by Biela et al describes a large scale investigation of the biophysical properties and structures of individual human tRNAs using cryoEM. The paper attempts to comprehensively survey the impact of 5 abundant pseudouridine modifications, and in the process of doing so, adds several important observations about the roles of individual modifications in biology. The use of cryoEM is very impressive given the size of the tRNA investigated, and they have laid the foundation to push this technique even further to answer important questions about tRNA biology and biochemistry. The paper is generally well-written and data are logically presented. I appreciate the effort to use consistent colors to keep the different sites of modification straight (with the exception of point 4 below). However, there are numerous examples of puzzling statements about the interpretation of the results or where potential confounding factors are completely overlooked, which together greatly diminish confidence in the results and would limit their overall impact. These are described in more detail below.

1) I do not see any evidence presented for functional PUS10 modification of the transcripts tested in this work in main text or supplemental figures, which contradicts the claims made in results (top of page 8) that the results show complete modification at all 5 naturally occurring pseudoU sites including 54, and again at the bottom of first paragraph citing fully modified tRNA. There is a very faint line right above the clear position for 55 modification (as would be expected for 54 modification) in figure 2C, but first, that band is also clearly seen in the lane 4 with inactive PUS10, as well as in PUS3, so it can not be convincingly attributed to 54. Second, this band is not reproducibly seen in any of the reactions shown in supplemental S2, which is in fact clearly known to the authors since they don't even cite it in the figure legend.

2) related to issue number 1, I am not surprised that PUS10 had no effect on any of the measured stability parameters because this is consistent with its clear lack of modification at this position. Although the text is correct that addition of the individual (or combined) PUS10 enzyme has no effect on most studied tRNAs, the reason for this can not be attributed to no effect of the modification because of the issue above. This should all be revised throughout this section on page 8-9, and throughout the rest of the paper and discussion, unless positive evidence for 54 modification can be obtained.

3) The use of the colored/gray dots to indicate combinations of modifications in Figure 2 is helpful, but should be extended to all plotted values corresponding to combinations of active/inactive mutants for better clarity

4) Figure 2D GluUUC, the dots above the PUS 4/7/10 lane do not seem to be correct—they are out of order with the legend order if I understand the way they were used in Figure 2D Gln UUG (should indicate PUS 4/7/10 from top to bottom, but the colors don't match this), and moreover they lane contains functional PUS10 but the dot appears to be gray. There seems to be the same issue with the dot color scheme in Figure 2E.

5) While I take the author's point that the use of inactive mutants is consistent with the effects on stability being due to the modifications themselves, I think the way this is written could be misleading. Particularly lines 174-175, page 8 is where I have this issue. Because of the way the tRNAs were purified after pseudoU addition by phenol/ethanol to remove any protein, any stabilizing effect of the enzymes binding to the tRNA can not be visualized anymore (the protein is gone by now). I appreciate the rigor of using catalytic mutants- they definitely rule out the possibility that inadvertently there were other modifications or changes to the tRNA made by any co-purifying activities that could have been the true source of the observed stability effects. I think this is a point well worth making. However, the stated point about not seeing stability effects due to tRNA binding could be easily misconstrued if someone doesn't understand that the enzymes are no longer there to exert (or not exert) any such effects. I believe this should be changed, both in results and in the discussion where this is proposed.

6) the data shown in Figure 4A do not seem to match the statements in the text about the individual effects of the modifications (on page 10-11) in the 2 points below:

a) I do not understand the data that are the basis for the statement on P.11 line 234 about Psi28 contributions to stability. the stability of tRNA^{His} with PUS1 is labeled as not significantly different from the unmodified, and moreover, there is no apparent significant difference between the tRNA treated with active vs. inactive PUS1 in the context of active PUS4/7. Unless there is some other support for this that I am missing, it does not seem to be a correct interpretation of the data.

b) for tRNA^{Lys}, the similarly increased stability observed with all three individual modifications (27, 39 and 55) seems to contradict the statement (line 236) that 39 and 55 play a less prominent role. [note typo here, I think the authors are referring to position 55, not 54, which is not present in this tRNA]

7) for Figure 4B, the statement about the improvement of overall resolution between modified and unmodified tRNA (lines 253-254) is obvious when comparing the unmodified to the doubly modified tRNAs shown in the middle panels. However, I can definitely not tell by eye alone whether this is also true for the tRNA in the right side of each figure (Gln with 13 and 55, and Gly with 13, 39 and 55). I don't see any significant difference in shading and if asked, I would have said there is overall lower resolution (more dark red) in both of these right side tRNAs relative to the unmodified. I would need to see some better quantification of this to support this statement.

8) I don't see what the box with all human nuclear modifications adds to Figure 5C, and may inadvertently suggest that all modifications are somehow present in this structure. I

would like to see this removed for clarity.

9) the authors mention, but do not ever discuss, the potential impact of the additional 2 5'-G nucleotides that they have incorporated into their in vitro transcripts. These nucleotides are not normally found on the 5'-end of tRNA, and given the defined structure of the aminoacyl-acceptor stem, the lack of an overall effect can not be assumed. Although there are no naturally known examples of tRNA with these additional nucleotides, it has been demonstrated for at least one tRNA (His) that the addition of excess 5'-nucleotides on the tRNA is sufficient to impact the biochemical properties and interaction of the tRNA with synthetases in vivo (Preston and Phizicky, RNA, 2010). There are several tested tRNAs (such as GlnUUG) that would potentially form base pairs with the 2 additional nucleotides (with U73 and C74), vs. others tested that would not predictably do so (ie those with A73). At the least, the authors should acknowledge and discuss the overall potential impact of this difference from native tRNA. This does not diminish the impact of the modifications overall, but may be impacting the absolute stabilities or other properties of the RNA near the 3' end.

Point-by-point response

Reviewer #1 (Remarks to the Author):

The authors present a large volume on experimental data on the influence of pseudouridylation by all human PUS enzymes on tRNA stability, especially the importance of these modifications to stabilize the elbow and junction regions, and that the modifications in these regions are mutually dependent. They use Tm measurements, cryo-EM and MD simulations to probe tRNA structure and dynamics. The new title is now specific to the study.

Major comments:

The biochemical data and simulations in the paper are elaborate and of substantial volume. The added simulations add value to the study. However, the paper would be a better one with less emphasis on the cryo-EM data. To draw conclusions about modifications and structural arrangements at the given resolution seems an over-interpretation of the data, also given the fact that better-resolved tRNAs are published in the context of larger complexes or the ribosome. In solution methods seem a better means to study the dynamics of molecules compared to cryo-EM.

Response: We appreciate the overall positive opinion about our work and we are glad that the reviewer agrees that the new title of our manuscript now better fits the focus of the presented study. In this respect, we would like to emphasize again, that obtaining high resolution structures of tRNAs bound to other complexes is not the main aim of the presented work – we have published these kind of studies for the Elongator complex¹⁻³, Pseudouridine Synthase 3⁴ and the ribosome⁵ – and we are currently preparing manuscripts for the already determined tRNA-bound cryo-EM structures at high resolutions of the CTU1/CTU2 complex, the FTSJ1-THADA complex, the FTSJ1-WDR6 complex and the QTRT1/2 complex. In the presented work, we provide data that opens the door for a complementary structural approach to study tRNAs in isolation – this main goal of our work has been recognized and appreciated by many leading scientists in the tRNA modification/processing field at various conference presentations and in numerous discussions over the last months – most recently at the 29th tRNA conference in Kanazawa, Japan. Our work paves the way for systematic analyses of the complex tRNA maturation process (e.g. processing, splicing and modifications) in the future. Due to the required high sample quantities, it is hard to imagine that such systematic structural analyses of numerous different stages of tRNA maturation would be feasible by using NMR or X-ray crystallography. However, we fully agree that for the analyses of specific events in-solution methods are/will be the better option. Therefore, we have undertaken additional efforts to remove any over-statements related to the used cryo-EM approach and over-interpretation of the data. We hope the reviewer agrees that the overall wealth of presented data from complementary techniques now justifies publication of our work in NSMB.

Line 151-153/ S5: Hence, we are confident that the quality of the reconstructions is sufficient to obtain reliable models of human tRNA. At the given resolution, all models of the different tRNAs will interchangeably fit into the maps. A good fit of a model is not necessarily an indicator of map quality.

Response: We agree with the reviewer and we have rephrased the sentence to state that it is not the quality of the map, but the combination of structural modeling together with restraints obtained from intermediate resolution cryo-EM that allows to obtain reliable models for human tRNA^{Gln_{UUG}} and tRNA^{Gly_{CCC}}. We have updated the phrase suggested by the reviewer, which now read as follows - “Hence, the combination of structural modeling together with spatial restraints from intermediate resolution cryo-EM maps is sufficient to obtain reliable models of human tRNA^{Gln_{UUG}} and tRNA^{Gly_{CCC}}.”.

Line 219: build would imply de-novo building. At the given resolution “place” would be appropriate to describe the rigid-body placement of a model (even with subsequent refinement).

Response: Changed as suggested.

Lines 223 following: While missing coulomb density often indicates flexibility in the molecule, I am not sure if the inverse is true. Replace “increased rigidity” by “reduced disorder”.

Response: Changed as suggested.

Line 228: “compaction of the elbow region” seems an overinterpretation at the given resolution: rather “featureless map towards recognizable tRNA features”.

Response: Changed as suggested.

Line 266: “clearly recognizable changes in the shape” better: would not lead to altered features of the coulomb density map...

Response: Changed as suggested.

Figure 3: The map-to-map comparison between different datasets, across different resolutions, with different particle numbers, is questionable at the given resolution for the relative size of the interpreted features. The plots in Figure S16 (per molecule) theoretically seem a more quantitative method to measure the disorder. But these plots fail to show the expected. In contrast to the main conclusion, Ψ 13 upper right panel has the most disorder in the direct neighborhood of position 13.

Response: In principle, we agree with the reviewer that the comparison between coulomb densities from different datasets would be questionable at the given resolution for the relative size of the interpreted features. However, we are pleased that the reviewer finds value in the newly presented per-molecule analyses (Figure S16), which provides a more quantitative analysis of conformational features upon the modification. We would like to highlight that the two neighboring residues of Ψ 13 are not affected by the disorder and our local stability analyses (Figure 4C) as well as our molecular simulations (Figure S17) show that the incorporation of Ψ s results in a very local stabilization effect. Ψ 13 base pairs with position 22 and it seems important to mention that the resolutions at positions 13 and 22 are relatively higher than that of neighboring nucleotides. In conclusion, when Ψ 13 is present in tRNA^{Gln}, the resolution at positions 13 and 22 remains comparable to that of neighboring nucleotides, even if the local resolution of positions 15-19 becomes lower. In our opinion, this illustrates the local effect of Ψ modifications on the structure. Therefore, the presented data are in line with the main conclusions and we assume that the additional disorder in the D-arm might be an indirect consequence of the compaction at the T-arm after modification of Ψ 55. The effect is not observed in tRNA^{Gln} carrying Ψ 13 and Ψ 39 (Figure 4B, middle panel) or tRNA^{Gly} (Figure S16C). Furthermore, we have previously analyzed the structural variation of the top 10 models within each ensemble individually and the increase in disorder in this region does not lead to strongly reduced Q-scores (Figure 3) or increased RMSD values (Figure S5) within the ensemble of the Ψ 13/ Ψ 55 modified tRNA^{Gln}.

Figure 4B: upper row, middle panel: here the compaction of the anticodon-stem is apparent through the better local resolution. All other panels lack clear meaning. It is not correct to compare local resolution across different samples, datasets, and data collections.

Response: Following the reviewer’s comment we have decided to move the comparison of tRNA^{Gly} to Supplementary Figure S16B. This reduction simplifies the figures and highlights the effect that has been confirmed by the reviewer. In addition, it strengthens the complementarity of Figure 4B with the local

stability analyses in Figure 4C, which is focusing on the effect of local stability in tRNA^{Gln}. The references and figure legends have been amended, accordingly.

Line 236: replace consistent with expected.

Response: This line has been removed following our most recent analyses of PUS10.

Line 275: rigidifies. Same issue as above, local resolution cannot be interpreted across datasets.

Response: Changed as suggested.

Minor comments:

Line 36: abundant instead of numerous

Response: The most “abundant” cellular RNA molecules are ribosomal RNAs – due to their large size, rRNA molecules contain the highest number RNA nucleotides (mass) per cell. As the number of individual tRNA molecules is much higher than the number of individual rRNA molecules, we here used “numerous” on purpose. We hope the reviewer agrees with our point of view that in this case “numerous” is the more appropriate term.

Line 46,47: define abbreviations at first use: D-arm T-arm.

Response: Abbreviations have been defined.

Line 97: remove “very” (subjective)

Response: Removed as suggested.

Line 108: Interesting, the species that do not fold properly might even absolutely rely on modifications for proper folding.

Response: We fully agree with the reviewer and we extended the sentence to mention this possibility.

S1: legend: occurring, no red circles in figure, free energy?

Response: We have updated the figure legend.

S1D: discrepancy in Glu? Why runs GG lower?

Response: Following the comment, we have repeated the analyses another time with fresh samples and the result remains the same. We assume that the additional stability after adding two Gs (Figure S1D) leads to a slight compaction of the molecules (even under denaturing conditions), which affects the running behavior. However, the exact reason for the surprising behavior remains unknown.

S2B: colour legend dots hard to see

Response: We have increased the size of the legends, including the dots.

S2D: Partially instead of Farcially

Response: We have corrected the typo.

S17A: legends to small

Response: We have increased the font size of the legends.

Figure 3: lower panel tRNAAspGUC?

Response: The typo has been corrected it in the figure and throughout the text.

Line 134: move specs to material and methods

Response: We have removed the specs of the microscope setup, which was already described in the material and methods section.

Line 173: define abbreviation at first time use

Response: Defined as suggested.

Line 188/189: ...strongly (remove) supporting.... Rest of the sentence is not clear to me

Response: We have removed the word “strongly” and we have rephrased the sentence to improve clarity. The sentence now reads as follows – “Incubation with a mixture of all inactive PUS enzymes shows no detectable Ψ modifications and no impact on the thermostability of tRNA^{Gln}_{UUU} (Figure 2D), supporting the specificity of the stabilizing effect by the Ψ modification itself.”

Figure 3: it would be nice to label the panels by letters and indicate these in the supplemental figures.

Response: Labels have been added as suggested. All references in the text and in the figure legends of the Supplemental figures have been updated.

Lines 250 following: Here, parts of the presented are speculations and could be moved to the discussion.

Response: Following the comment by the reviewer and the additional issues of reviewer 3 concerning PUS10, we have moved the section to the discussion, where we also discuss newly obtained data for human PUS10.

line 272: remove changes

Response: Removed as suggested

line 371: remove the bold claim line 371: remove the bold claim, <https://doi.org/10.1073/pnas.2320493121> has similarly sized particles

Response: We are very much aware of the mentioned manuscript, as we have published similar work on the 5' region of beta-coronaviruses at the same time⁷. The constructs used for the cryo-EM analyses of the viral SL5 regions are around 120 nucleotides and therefore almost double the size of tRNAs. In our most recent publication in *Nature Protocols* with Zhaoming Su's group⁶, we provide a summary of all available RNA-only cryo-EM structures. In accordance to our analysis, our work on individual tRNAs does represent the smallest asymmetric RNA molecules analysed by cryo-EM to date. However, we agree that the HIV-1 RNA dimerization signal⁸ and the frameshift stimulation element in the SARS-CoV-2 RNA genome⁹ are of almost similar size. Therefore, we rephrased the claim to “Our cryo-EM work on individual tRNAs – representing one of the smallest asymmetric RNA particles analysed by cryo-EM to date - ...”.

Reviewer #2 (Remarks to the Author):

The authors very thoroughly addressed all the points I raised. They added new data and clarified several sentences. Overall, this is an important paper contributing to our fine understanding of the tRNA molecular properties and their impacts on tRNA function and translation.

Response: We highly appreciate the positive opinion by the reviewer.

Reviewer #3 (Remarks to the Author):

The revised manuscript including significant new and reanalyzed data is substantially improved. There are many positives in this paper that were previously enumerated, and most of the issues raised in earlier review were substantively and appropriately addressed. However, the new data provided for PUS10, while an improvement on the previous results, are still not sufficiently convincing.

Response: We appreciate that the reviewer recognized the substantial improvement in the revised version of manuscript and that most of the previously raised issues were resolved. The detailed analysis of PUS10 has indeed been challenging, but we would like to highlight that we clearly detect its modification activity *in vitro*. Following the reviewer's suggestion, we managed to establish experimental conditions that allow to discriminate between Ψ on position 54 and 55 in tRNA^{Gln} using CMC-based primer extension assays. Strikingly, we observed that the target specificity of purified human PUS10 is far less stringent than of the other four tested human stand-alone PUS enzymes *in vitro*. In detail, we observed that purified PUS10 can install Ψ at two positions, namely 54 and 55, in tRNA^{Gln}. Even if the signal for both positions appears similar, we conclude that human PUS10 still prefers 54, due to a reduced read-through efficiency at Ψ 55. As PUS10 clearly shows activity towards position 54 and 55, it is indeed difficult to dissect the specific contribution of Ψ 54 on the stability of all tested tRNA(s). Of note, this observation coincides with a previous report that archaeal PUS10 modifies both positions in the T-arm¹⁰. After the absolutely correct statement of the reviewer (in combination with critical reports in the literature), we also realized that our current data suggests that tRNA^{His} is simply not a target for PUS10 at all. Following these results, we agree with the reviewer that the data obtained for PUS10 on tRNA^{Glu} and tRNA^{His} should be excluded from the manuscript. As the data represents one of the first biochemical characterizations of purified human PUS10 and its target specificity (Figure S6C), we believe that the analyses of PUS10 *in vitro* activity on tRNA^{Gln} should remain in the manuscript (Figure 2C). Due to the ambiguity for PUS10-mediated pseudouridylation on both positions, we nevertheless have excluded all T_m value analyses for PUS10 on tRNA^{Gln} as well. Furthermore, we have remeasured T_m values for all samples of tRNA^{Gln}, tRNA^{Glu} and tRNA^{His} that contained PUS10 in combination with other PUS enzymes in the absence of PUS10.

We would like to thank the reviewer for insisting on these analyses and we believe that our additional data now clarify the observed effects and provide additional insights into the *in vitro* activity of PUS10. We have updated all affected figures and text, including discussion, in the revised manuscript. We hope the reviewer agrees that the overall wealth of presented data from complementary techniques now justifies publication of our work in NSMB.

1) because of the limited resolution in the region of the gel close to the primer, the difference between 54 vs. 55 modification is not clear, and questions remain about the extent of 54 modification even in the best (Gln) substrate, and this is also an issue for tRNA^{Glu}. This is particularly an issue in the lanes where PUS4 and PUS10 are combined- there should be single nucleotide resolution of each band to confirm that both modifications are present at substantial amounts. Perhaps running a higher percentage gel and running samples with the two modifications side by side could help to address this issue and provide convincing evidence for the presence of the 54 modification in substantial amounts, as suggested.

Response: The detection of the neighboring modification is technically very challenging. In particular, as we are limited by the positioning of the primer for detecting changes at these positions close to the 3' end of tRNAs. Following the suggestion we have nonetheless undertaken additional efforts to confirm the presence of both modifications by running higher percentage gels and running tRNA^{Gln} samples with the two modifications side by side (see new Figure S6C). As described above, we found that human PUS10 can introduce Ψ at the two neighboring positions, namely 54 and 55, in tRNA^{Gln}. This observation may

also explain why we have seen a stabilization of tRNA^{Gln} by PUS10, which was most likely caused by the presence of Ψ55 in tRNA^{Gln} after the treatment with PUS10. Purified PUS4 shows high selectivity for 55 and in our data there is no indication that it modifies position 54. Moreover, we have tested whether the order of performing the modification reactions by PUS4 and PUS10 has an influence on the outcome of the modification pattern. The result shows that we can still detect both, Ψ54 and Ψ55, when we treated tRNA^{Gln} with PUS10 before PUS4. In this sequential reaction, we see an additional increase of Ψ55 in comparison to the PUS10 reaction alone, which suggests that only a small fraction of Ψ55 is modified by PUS10 even after 8 hours of incubation. In contrast, we detected a weaker signal of Ψ54, when the treatment order is reversed, hinting that the amount of Ψ55 is significantly higher in this condition. Due to the reduced read-through efficiency by the efficient PUS4-dependent installation of Ψ55 in these primer extension assay, it is impossible to quantify the readout if PUS10-dependent Ψ54 levels are affected by the preceding modification by PUS4. As any of these additional conclusions appears too speculative at the moment, we only included the result showing PUS10 activity on tRNA^{Gln} (Figure S6C), but excluded all PUS10 related tRNA stabilization data from the manuscript.

2) The inclusion of the PUS10 results for tRNA^{His} (figure 4) are unjustified and not supported by the data presented. The authors have shown no convincing in vitro evidence for modification of their tRNA at this position (especially in reviewer figure 4, and even in Figure S15 where a very faint band that is close to background and not as robust as the possible position 55 modification is suggested. Also, the position of this band appears to be slightly below the migration of the 55 band, which is the opposite of what would be expected for this position- see comment #1 above about the resolution issue). Therefore, the implication that any role for 54 modification on stability or structure of this tRNA^{His} can be ascertained is not justified by the data. Unless data can be presented that convincingly show this modification is happening, the impact and analysis of this modification should be removed from all figures and text for this tRNA. The discussion on p. 12 about the questionable modification status and their observation of weak/no ability to modify this tRNA in vitro could remain.

Response: We fully agree with the reviewer and we have removed all data that relate to PUS10 modifications in tRNA^{His}. We have added the analysis of tRNA^{His} after modification by a combination of PUS1/4/7 and included the newly generated data in the revised Figure 4A and Figure S6. Following the suggestion by reviewer 1, we have moved the discussion about PUS10 (previously on page 12) from result section into the discussion section. Furthermore, we extended the paragraph with comments and the reasoning for not introducing PUS10 for these samples due to the inability to independently dissect the impact of Ψ54 *in vitro*.

3) For tRNA^{Glu}, although there is a more convincing suggestion based on the activity seen in the reviewer figure and figure S6, the resolution issue remains. I am also concerned about the apparently dramatic decrease in stability that is implied to be a result of PUS10 modification of Glu, since this was also seen in tRNA^{His} to some degree, despite the fact that there does not seem to be any significant modification at 54. These results are sufficiently ambiguous that I would like to see additional quantification of the mutation using some complementary method to really get at whether there is enough modification at this position that it could reasonably be expected to be the cause of the observed effect.

Response: We agree with the reviewer on the resolution issue and have made efforts to improve it and indeed we can observe PUS10 activity towards Ψ54 than Ψ55 in tRNA^{Gln} (see responses above and Figure S6C). As it is difficult to analyze the impact of Ψ54 alone on all tested tRNAs, we have removed these stability analyses of PUS10 on tRNA^{His}, tRNA^{Gln} and tRNA^{Glu} from the manuscript. We believe this change makes the manuscript concise and focusing on specific Ψ, including Ψ13, Ψ27/28, Ψ39 and Ψ55, in tRNAs. Of note, none of other analyses and main conclusions of the manuscript are affected by the removal of the PUS10-related data.

Literature

1. Dauden, M. I. *et al.* Molecular basis of tRNA recognition by the Elongator complex. *Sci Adv* **5**, (2019).
2. Jaciuk, M. *et al.* Cryo-EM structure of the fully assembled Elongator complex. *Nucleic Acids Res* **51**, 2011–2032 (2023).
3. Abbassi, N.-E.-H. *et al.* Cryo-EM structures of the human Elongator complex at work. *Nat Commun* **15**, 4094 (2024).
4. Lin, T.-Y. *et al.* The molecular basis of tRNA selectivity by human pseudouridine synthase 3. *Mol Cell* **84**, 2472–2489.e8 (2024).
5. Jain, S. *et al.* Modulation of translational decoding by m6A modification of mRNA. *Nat Commun* **14**, 4784 (2023).
6. Chen, X. *et al.* RNA sample optimization for cryo-EM analysis. *Nat Protoc* (2024) doi:10.1038/s41596-024-01072-1.
7. de Moura, T. R. *et al.* Conserved structures and dynamics in 5'-proximal regions of Betacoronavirus RNA genomes. *Nucleic Acids Res* **52**, 3419–3432 (2024).
8. Zhang, K. *et al.* Structure of the 30 kDa HIV-1 RNA Dimerization Signal by a Hybrid Cryo-EM, NMR, and Molecular Dynamics Approach. *Structure* **26**, 490–498.e3 (2018).
9. Zhang, K. *et al.* Cryo-EM and antisense targeting of the 28-kDa frameshift stimulation element from the SARS-CoV-2 RNA genome. *Nat Struct Mol Biol* **28**, 747–754 (2021).
10. Gurha, P. & Gupta, R. Archaeal Pus10 proteins can produce both pseudouridine 54 and 55 in tRNA. *Rna* **14**, 2521–2527 (2008).

Dear Dr Glatt,

thank you for submitting a revised version of your manuscript. Your study has now been seen by an arbitrating referee, who finds that the concerns raised by the previous referees have been addressed and recommends publication of the manuscript with a couple of minor textual edits which you can find below. In addition there remain a few mainly editorial points that have to be addressed before I can extend formal acceptance of the manuscript:

- Please double-check to make sure to all relevant funding information in the manuscript is also entered into our submission system. (Missing in the system currently: Polish high-performance computing infrastructure PLGrid (HPC Centers: ACK Cyfronet AGH) grants PLG/2020/014021, PLG/2021/014959 and PLG/2022/015771 (ADB), and PLG/2023/016901)

- Please reduce the number of keywords on the abstract page to five (ideally choosing broad general terms).

- Please adjust the format of the reference list and of the in-text citations according to EMBO Journal format (alphabetical order, author name et al + year.../up to 10 author names in the reference list before et al / please refer to our Guide to Authors for additional information on EMBO J reference format).

- Please rename the Conflict of Interest section into "Disclosure and Competing Interests Statement", in accordance with our updated Guide to Authors (<https://www.embopress.org/competing-interests>)

- As we are switching from a free-text author contribution statement towards a more formal statement based on Contributor Role Taxonomy (CRediT) terms, please remove the present Author Contribution section and instead specify each author's contribution(s) directly in the Author Information page of our submission system during upload of the final manuscript. See <https://casrai.org/credit/> for more information.

- Please adjust the in-text callouts for individual figures and figure panels: e.g there is a callout for Table 1 and no such table uploaded and there are also missing callouts for Fig. 3A-D

- Please provide either a "Yes" or a "Not Applicable" answer to each one of the questions in your Author Checklist (<https://www.embopress.org/pb-assets/embo-site/EMBO%20Press%20Author%20Checklist-1642513524327.xlsx>). In the last column of this checklist, only the sections of the manuscript where the relevant information can be found should be listed (the information per se should be included in the main manuscript file).

- Please remove the figures need from the ms file and only figure legends should remain in ms placed below the References

- DATASET EV LEGENDS: nomenclature should be Dataset EV1; legend should be removed from Appendix PDF and included as a separate tab/sheet in the Excel file

APPENDIX 1 FILE WITH ToC: Appendix file needs to be in PDF format; title page should contain "Appendix for + ms title" and ToC with the page numbers for the listed items; nomenclature should be Appendix Figure Sx and Appendix Table Sx throughout ms and Appendix PDF; references should be alphabetical with 10 authors + et al.

- Please provide the Reagent and Tools Table. For more information, please check <https://www.embopress.org/page/journal/14602075/authorguide#structuredmethods> and download the template for Reagent Table

- Please provide source data. My colleague Hannah Sonntag will send a official source data request with additional information in a separate e-mail.

- Please provide suggestions for a short 'blurb' text prefacing and summing up the conceptual aspect of the study in two sentences (max. 250 characters), followed by 3-5 one-sentence 'bullet points' with brief factual statements of key results of the paper; they will form the basis of an editor-written 'Synopsis' accompanying the online version of the article. Please also provide an altered synopsis image, making sure that the aspect ratio conforms to our website's format - it should be exactly 550 pixels wide and between 300-600 pixels high.

- There is a potential Image re-use between Appendix Fig S7 and S8. Same images even though the figure legend says one is Pus 4/7 modified and the other is Pus 3/7 modified. Please doublecheck these figures. A short explanation would be helpful.

- Please provide the specific URLs for EMPIAR-11514, EMD-16940; EMD-16939; EMPIAR-11515, EMD-16941; EMPIAR 11516, EMD-16943; EMPIAR-11517, EMD-16945; EMD-16946; EMD-16947; EMPIAR-11518, EMD-16948, EMPIAR-11519, EMD-16951 datasets in the data availability statement.

- Figure Legends (main + EV): 1. Please note that the exact p values are not provided in the legends of figures 2A, D, E, F, G,

H; 4A

2. Please indicate what */ **/ ***/ **** represents; if this represents p value(s), please indicate the statistical test used and where appropriate, specify the exact p value in the legend(s) of figure(s) 4C

3. Please note that the box plots need to be defined in terms of minima, maxima, centre, bounds of box and whiskers, and percentile in the legends of figures 2A, D, E, F, G, H; 4A, C

4. Please note that information related to n is missing in the legends of figures 4A, C

- Please rename the movie files to Movie EV1-EV4 with the corresponding callouts, and the legends should be removed from Appendix PDF and zipped with each movie file.

- Please adjust the order of the manuscript sections: Title page with complete author information, Abstract, Keywords, Introduction, Results, Discussion, Methods, Data Availability Section, Acknowledgements, Disclosure and Competing Interests Statement, References, Main figure legends, Tables, Expanded Figure Legends.

With best regards,

Cornelius Schneider

Cornelius Schneider, PhD
Editor | The EMBO Journal
c.schneider@embojournal.org

- a point-by-point response to the referees' comments, with a detailed description of the changes made (as a word file).

- a word file of the manuscript text.

- individual production quality figure files (one file per figure)

- a complete author checklist, which you can download from our author guidelines

(<https://www.embopress.org/page/journal/14602075/authorguide>).

- Expanded View files (replacing Supplementary Information)

- a Reagents and Tools Table as part of the Methods section, which can be downloaded from our author guidelines

(<https://www.embopress.org/page/journal/14602075/authorguide#structuredmethods>)

We realize that it is difficult to revise to a specific deadline. In the interest of protecting the conceptual advance provided by the work, we recommend a revision within 3 months (8th Jun 2025). Please discuss the revision progress ahead of this time with the editor if you require more time to complete the revisions. Use the link below to submit your revision:

Referee #1:

I have carefully reviewed the revised version of the manuscript "Determining the effects of Pseudouridine incorporation on human tRNAs by single particle cryo-EM, biophysics, and computational analyses" by Biela et al., as well as the authors' point-by-point responses. The authors have adequately addressed the concerns raised in the initial review.

This study presents a systematic and integrative analysis of the effects of pseudouridylation (Ψ) on human tRNA stability using cryo-EM, molecular dynamics simulations, and biophysical approaches. The results show that specific Ψ sites, particularly Ψ 13 and Ψ 55, stabilize tRNA structure by reinforcing interactions between the D-arm and T-arm. By successfully resolving intermediate-resolution cryo-EM structures of isolated tRNAs, the authors demonstrate that single-particle cryo-EM can be applied to studying small RNA molecules. Given the growing interest in RNA modifications and their impact on translation, tRNA stability, and cellular regulation, this work is timely and relevant for researchers in the field or neighbouring fields.

However, a few remaining conceptual and methodological considerations should be addressed in order to ensure that the manuscript presents a balanced discussion of its findings and limitations.

Points for a more balanced discussion

1. Over-Reliance on Melting Temperature (T_m) as a Stability Metric

While T_m is a useful parameter for assessing RNA stability, it is a relatively coarse measurement that does not capture finer aspects of folding kinetics, local flexibility, or intermediate states. The study could benefit from discussing how other biophysical approaches (e.g. FRET) might provide additional insights into tRNA stability. This is particularly relevant for cases where T_m shifts are relatively small, as minor differences may not be functionally meaningful.

2. Lack of Direct Structural Visualization of Ψ

The cryo-EM part of the study shows that Ψ modifications influence tRNA stability and structure. Still, because cryo-EM does not directly resolve modifications like Ψ , their effects are inferred from indirect features such as backbone rigidity and local resolution improvements. The authors should explicitly acknowledge this limitation in the Discussion section.

3. Potential Impact of Other tRNA Modifications

While Ψ is an important tRNA modification, it does not act in isolation. Many other post-transcriptional modifications are known to influence tRNA structure and function. The study focuses exclusively on Ψ , but the authors should briefly discuss the role of other modifications and acknowledge that the absence of these modifications in *in vitro* transcribed tRNAs (IVT tRNAs) may influence their observations. *In vivo*, Ψ likely acts in combination with these modifications rather than being the sole determinant of stability. Importantly, because the study focuses exclusively on pseudouridylation, it cannot establish the relative importance of Ψ compared to other modifications. The observed stability effects could, in part, be due to the absence of other stabilizing modifications rather than the presence of Ψ alone. The authors should explicitly state that this study does not evaluate the full range of tRNA modifications contributing to folding and stability.

4. Limited Structural Diversity in Studied tRNAs

The analyzed tRNAs belong to well-structured canonical classes, and the observed stabilization effects may not generalize to structurally atypical tRNAs, such as mitochondrial tRNAs, which often lack complete D- or T-arms. The authors should acknowledge this limitation and discuss whether Ψ modifications would affect non-canonical tRNA structures differently.

5. Biological Relevance of Endogenous tRNA Structural Studies

While the authors successfully isolated and analyzed endogenous tRNAs, their cryo-EM reconstructions were limited to ~ 10 Å resolution, restricting detailed structural interpretation. The manuscript would benefit from discussing potential ways to improve resolution in future endogenous tRNA studies.

Minor points in the current text:

1. Specification of Reagents in Online Methods (Page 31): The manuscript does not state the manufacturers of RNasin, T7 RNA polymerase, or pyrophosphatase. Providing supplier information ensures experimental reproducibility.

2. Details on Pseudouridine Synthase (PUS) Expression and Purification (Page 32): The description of PUS enzyme production is too general. Each enzyme likely required distinct expression and purification conditions, and these details should be included for reproducibility.
3. Source of HEK 293T Cells (Page 34): The manuscript states that HEK 293T cells were obtained from GE Healthcare, but GE Healthcare is not a known supplier of cell lines. The authors should clarify whether these were obtained from Dharmacon (a GE subsidiary), ATCC, or another supplier.
4. Description of APM Gel Electrophoresis for Thiolated tRNAs: The manuscript mentions that APM gels were used to confirm the presence of thiolated tRNAs, but no details are provided on the protocol. The authors should describe the gel composition, running conditions, and how the migration shift was interpreted.
5. Clarification of the RiboGreen{trade mark, serif} Assay Mechanism (Page 6): The fluorescence of RiboGreen{trade mark, serif} increases as tRNA unfolds, allowing more single-stranded regions to become accessible. The current wording does not make this mechanism clear and should be refined. Please consider "by monitoring the increase in fluorescence intensity as RiboGreen{trade mark, serif} binds to tRNA regions that become single-stranded upon unfolding under a temperature gradient." or similar.
6. Correction of Cryo-EM Terminology (Page 13): The manuscript refers to a "Coulomb density map," but in cryo-EM, the correct term is "Coulomb potential map". This should be updated for accuracy.

Point-by-point response

Reviewer #4 arbitrating referee

I have carefully reviewed the revised version of the manuscript "Determining the effects of Pseudouridine incorporation on human tRNAs by single particle cryo-EM, biophysics, and computational analyses" by Biela et al., as well as the authors' point-by-point responses. The authors have adequately addressed the concerns raised in the initial review.

This study presents a systematic and integrative analysis of the effects of pseudouridylation (Ψ) on human tRNA stability using cryo-EM, molecular dynamics simulations, and biophysical approaches. The results show that specific Ψ sites, particularly Ψ 13 and Ψ 55, stabilize tRNA structure by reinforcing interactions between the D-arm and T-arm. By successfully resolving intermediate-resolution cryo-EM structures of isolated tRNAs, the authors demonstrate that single-particle cryo-EM can be applied to studying small RNA molecules. Given the growing interest in RNA modifications and their impact on translation, tRNA stability, and cellular regulation, this work is timely and relevant for researchers in the field or neighbouring fields.

However, a few remaining conceptual and methodological considerations should be addressed in order to ensure that the manuscript presents a balanced discussion of its findings and limitations.

Response: We thank the reviewer for the appreciation of our work and the fair and constructive comments. We have revised the manuscript to balance the discussion and also highlight the limitation of our study.

Points for a more balanced discussion

1. Over-Reliance on Melting Temperature (T_m) as a Stability Metric

While T_m is a useful parameter for assessing RNA stability, it is a relatively coarse measurement that does not capture finer aspects of folding kinetics, local flexibility, or intermediate states. The study could benefit from discussing how other biophysical approaches (e.g. FRET) might provide additional insights into tRNA stability. This is particularly relevant for cases where T_m shifts are relatively small, as minor differences may not be functionally meaningful.

Response: Done as suggested. We added a section to the discussion section on page 17 to acknowledge the other methods and discuss the added value of combining them with our approach in the future. The section reads as follows – “*Our findings demonstrate that T_m values report on global as well as local stabilizing effects caused by introducing Ψ into tRNAs. However, this relatively coarse measurement that does not capture finer aspects of folding kinetics, local flexibility, or intermediate states. Therefore, this measurement needs to be combined with complementary approaches (e.g. FRET, NMR, AFM) to further enhance our understanding of tRNA folding and dynamics.*”

2. Lack of Direct Structural Visualization of Ψ

The cryo-EM part of the study shows that Ψ modifications influence tRNA stability and structure. Still, because cryo-EM does not directly resolve modifications like Ψ , their effects are inferred from indirect features such as backbone rigidity and local resolution improvements. The authors should explicitly acknowledge this limitation in the Discussion section.

Response: Done as suggested. We expanded the discussion section on page 16 to acknowledge this limitation of our study. The section reads as follows – “*We would like to mention that due to the limited resolution of our cryo-EM reconstructions, we are not able to directly observe the conformation of the introduced Ψ s. Using molecular dynamics simulations, we provide a mechanistic explanation for the observed stabilization, which we inferred indirectly from local resolution improvements.*”

3. Potential Impact of Other tRNA Modifications

While Ψ is an important tRNA modification, it does not act in isolation. Many other post-transcriptional modifications are known to influence tRNA structure and function. The study focuses exclusively on Ψ , but the authors should briefly discuss the role of other modifications and acknowledge that the absence of these modifications in in vitro transcribed tRNAs (IVT tRNAs) may influence their observations. In vivo, Ψ likely acts in combination with these modifications rather than being the sole determinant of stability. Importantly, because the study focuses exclusively on pseudouridylation, it cannot establish the relative importance of Ψ compared to other modifications. The observed stability effects could, in part, be due to the absence of other stabilizing modifications rather than the presence of Ψ alone. The authors should explicitly state that this study does not evaluate the full range of tRNA modifications contributing to folding and stability.

Response: We fully agree with the reviewer and in an earlier version of the manuscript, Figure 5 contained a complete list of other modifications to highlight the issue mentioned by the reviewer. Upon the request of another reviewer, we had to remove this list of other modification, because they were considered as distracting from the results of our study. Following the comment, we expanded the results section on page 16. The paragraph acknowledges the raised issue as follows – “*In addition, our study focuses exclusively on Ψ s and the used in vitro transcribed tRNAs lack other modifications. Hence, future studies need to include the full range of tRNA modification to understand how they contribute to folding and stability of tRNAs and how they affect the stabilizing effects of Ψ s, determined in the study.*”

4. Limited Structural Diversity in Studied tRNAs

The analyzed tRNAs belong to well-structured canonical classes, and the observed stabilization effects may not generalize to structurally atypical tRNAs, such as mitochondrial tRNAs, which

often lack complete D- or T-arms. The authors should acknowledge this limitation and discuss whether Ψ modifications would affect non-canonical tRNA structures differently.

Response: We fully agree. We have mentioned atypical tRNAs, like armless tRNAs in the discussion on page 19 and included a suitable reference. We have now elaborated on this topic and rephrased the term to “*armless tRNAs present in the mitochondria of certain species*”. In addition, we added a sentence to address the issue raised, which reads as follows – “*In addition, it remains to be shown, whether Ψ plays a similar role for the folding and stability of these atypical tRNAs.*”

5. Biological Relevance of Endogenous tRNA Structural Studies

While the authors successfully isolated and analyzed endogenous tRNAs, their cryo-EM reconstructions were limited to ~ 10 Å resolution, restricting detailed structural interpretation. The manuscript would benefit from discussing potential ways to improve resolution in future endogenous tRNA studies.

Response: Agreed. We have added a short sentence about how to improve the cryo-EM analyses of endogenous tRNAs in the future. The section reads as follows – “*Future studies will need to focus on solving these issues by optimizing purification protocols for each individual tRNA, by customizing sample preparation methods, by further improving cryo-EM data collection strategies and data analyses pipelines.*”

Minor points in the current text:

1. Specification of Reagents in Online Methods (Page 31): The manuscript does not state the manufacturers of RNasin, T7 RNA polymerase, or pyrophosphatase. Providing supplier information ensures experimental reproducibility.

Response: Done as suggested. Supplier details have been added.

2. Details on Pseudouridine Synthase (PUS) Expression and Purification (Page 32): The description of PUS enzyme production is too general. Each enzyme likely required distinct expression and purification conditions, and these details should be included for reproducibility.

Response: Done as suggested. Details about expression and purification conditions have been added.

3. Source of HEK 293T Cells (Page 34): The manuscript states that HEK 293T cells were obtained from GE Healthcare, but GE Healthcare is not a known supplier of cell lines. The authors should clarify whether these were obtained from Dharmacon (a GE subsidiary), ATCC, or another supplier.

Response: Described as suggested. Cells were indeed obtained from Dharmacon, a subsidiary of GE Healthcare. The information has been updated.

4. Description of APM Gel Electrophoresis for Thiolated tRNAs: The manuscript mentions that APM gels were used to confirm the presence of thiolated tRNAs, but no details are provided on the protocol. The authors should describe the gel composition, running conditions, and how the migration shift was interpreted.

Response: Described as suggested. Additional technical details have been added for the APM Gel Electrophoresis in the Methods section.

5. Clarification of the RiboGreen™ Assay Mechanism (Page 6): The fluorescence of RiboGreen™ increases as tRNA unfolds, allowing more single-stranded regions to become accessible. The current wording does not make this mechanism clear and should be refined. Please consider "by monitoring the increase in fluorescence intensity as RiboGreen™ binds to tRNA regions that become single-stranded upon unfolding under a temperature gradient." or similar.

Response: Done as suggested. We have change the sentence on page 6 in accordance to the suggestion.

6. Correction of Cryo-EM Terminology (Page 13): The manuscript refers to a "Coulomb density map," but in cryo-EM, the correct term is "Coulomb potential map". This should be updated for accuracy.

Response: We have changed the phrase on page 13, accordingly.

Dear Dr. Glatt,

I am pleased to inform you that your manuscript has been accepted for publication in the EMBO Journal.

Yours sincerely,

Cornelius Schneider, PhD
Editor
The EMBO Journal
c.schneider@embojournal.org
